# Generalization through the Lens of Leave-one-out Error

Gregor Bachmann [a], Thomas Hofmann[a], and Aurélien Lucchi[b]

[a]Department of Computer Science, ETH Zürich, Switzerland
[b]Department of Mathematics and Computer Science, University of Basel ,
{gregor.bachmann, thomas.hofmann}@inf.ethz.ch
aurelien.lucchi@unibas.ch

## Abstract

Despite the tremendous empirical success of deep learning models to solve various learning tasks, our theoretical understanding of their generalization ability is very limited. Classical generalization bounds based on tools such as the VC dimension or Rademacher complexity, are so far unsuitable for deep models and it is doubtful that these techniques can yield tight bounds even in the most idealistic settings (Nagarajan & Kolter, 2019). In this work, we instead revisit the concept of leave-one-out (LOO) error to measure the generalization ability of deep models in the so-called kernel regime. While popular in statistics, the LOO error has been largely overlooked in the context of deep learning. By building upon the recently established connection between neural networks and kernel learning, we leverage the closed-form expression for the leave-one-out error, giving us access to an efficient proxy for the test error. We show both theoretically and empirically that the leave-one-out error is capable of capturing various phenomena in generalization theory, such as double descent, random labels or transfer learning. Our work therefore demonstrates that the leave-one-out error provides a tractable way to estimate the generalization ability of deep neural networks in the kernel regime, opening the door to potential, new research directions in the field of generalization.

## 1 Introduction

Neural networks have achieved astonishing performance across many learning tasks such as in computer vision (He et al., 2016), natural language processing (Devlin et al., 2019) and graph learning (Kipf & Welling, 2017) among many others. Despite the large overparametrized nature of these models, they have shown a surprising ability to generalize well to unseen data. The theoretical understanding of this generalization ability has been an active area of research where contributions have been made using diverse analytical tools (Arora et al., 2018; Bartlett et al., 2019; 2017; Neyshabur et al., 2015; 2018). Yet, the theoretical bounds derived by these approaches typically suffer from one or several of the following limitations: i) they are known to be loose or even vacuous (Jiang et al., 2019), ii) they only apply to randomized networks (Dziugaite & Roy, 2018; 2017), and iii) they rely on the concept of uniform convergence which was shown to be non-robust against adversarial perturbations (Nagarajan & Kolter, 2019).

In this work, we revisit the concept of leave-one-out error (LOO) and demonstrate its surprising ability to predict generalization for deep neural networks in the so-called kernel regime. This object is an important statistical estimator of the generalization performance of an algorithm that is theoretically motivated by a connection to the concept of uniform stability (Pontil, 2002). It has also recently been advocated by Nagarajan & Kolter (2019) as a potential substitute for uniform convergence bounds. Despite being popular in classical statistics (Cawley & Talbot, 2003; Vehtari et al., 2016; Fukunaga & Hummels, 1989; Zhang, 2003), LOO has largely been overlooked in the context of deep learning, likely due to its high computational cost. However, recent advances from the neural tangent kernel perspective (Jacot et al., 2018) render the LOO error all of a sudden tractable thanks to the availability of a closed-form expression due to Stone (1974, Eq. 3.13). While the role of the LOO error as an estimator of the generalization performance is debated in the statistics community

(Zhang & Yang, 2015; Bengio & Grandvalet, 2004; Breiman, 1996; Kearns & Ron, 1999), the recent work by Patil et al. (2021) established a consistency result in the case of ridge regression, ensuring that LOO converges to the generalization error in the large sample limit, under mild assumptions on the data distribution.

Inspired by these recent advances, in this work, we investigate the use of LOO as a generalization measure for deep neural networks in the kernel regime. We find that LOO is a surprisingly rich descriptor of the generalization error, capturing a wide range of phenomena such as random label fitting, double descent and transfer learning. Specifically, we make the following contributions:

- We extend the LOO expression for the multi-class setting to the case of zero regularization and derive a new closed-form formula for the resulting LOO accuracy.
- We investigate both the LOO loss and accuracy in the context of deep learning through the lens of the NTK, and demonstrate empirically that they both capture the generalization ability of networks in the kernel regime in a variety of settings.
- We showcase the utility of LOO for practical networks by accurately predicting their transfer learning performance.
- We build on the mathematically convenient form of the LOO loss to derive some novel insights into double descent and the role of regularization.

Our work is structured as follows. We give an overview of related works in Section 2. In Section 3 we introduce the setting along with the LOO error and showcase our extensions to the multi-class case and LOO accuracy. Then, in Section 4 we analyze the predictive power of LOO in various settings and present our theoretical results on double descent. We discuss our findings and future work in Section 5. We release the code for our numerical experiments on *Github*[1].

## 2 RELATED WORK

Originally introduced by Lachenbruch (1967); Mosteller & Tukey (1968), the leave-one-out error is a standard tool in the field of statistics that has been used to study and improve a variety of models, ranging from support vector machines (Weston, 1999), Fisher discriminant analysis (Cawley & Talbot, 2003), linear regression (Hastie et al., 2020; Patil et al., 2021) to non-parametric density estimation (Fukunaga & Hummels, 1989). Various theoretical perspectives have justified the use of the LOO error, including for instance the framework of uniform stability (Bousquet & Elisseeff, 2002) or generalization bounds for KNN models (Devroye & Wagner, 1979). Elisseeff et al. (2003) focused on the stability of a learning algorithm to demonstrate a formal link between LOO error and generalization. The vanilla definition of the LOO requires access to a large number of trained models which is typically computationally expensive. Crucially, Stone (1974) derived an elegant closed-form formula for kernels, completely removing the need for training multiple models. In the context of deep learning, the LOO error remains largely unexplored. Shekkizhar & Ortega (2020) propose to fit polytope interpolators to a given neural network and estimate its resulting LOO error. Alaa & van der Schaar (2020) rely on a Jackknife estimator, applied in a LOO fashion to obtain better confidence intervals for Bayesian neural networks. Finally, we want to highlight the concurrent work of Zhang & Zhang (2021) who analyze the related concept of influence functions in the context of deep models.

Recent works have established a connection between kernel regression and deep learning. Surprisingly, it is shown that an infinitely-wide, fully-connected network behaves like a kernel both at initialization (Neal, 1996; Lee et al., 2018; de G. Matthews et al., 2018) as well as during gradient flow training (Jacot et al., 2018). Follow-up works have extended this result to the non-asymptotic setting (Arora et al., 2019a; Lee et al., 2019), proving that wide enough networks trained with small learning rates are in the so-called kernel regime, i.e. essentially evolving like their corresponding tangent kernel. Moreover, the analysis has also been adapted to various architectures (Arora et al., 2019a; Huang et al., 2020; Du et al., 2019; Hron et al., 2020; Yang, 2020) as well as discrete gradient descent (Lee et al., 2019). The direct connection to the field of kernel regression enables the usage of the closed-form expression for LOO error in the context of deep learning, which to the best of our knowledge has not been explored yet.

---

[1]https://github.com/gregorbachmann/LeaveOneOut

## 3 SETTING AND BACKGROUND

In the following, we establish the notations required to describe the problem of interest and the setting we consider. We study the standard learning setting, where we are given a dataset of input-target pairs $\mathcal{S} = \{(\boldsymbol{x}_1, \boldsymbol{y}_1), \ldots, (\boldsymbol{x}_n, \boldsymbol{y}_n)\}$ where each $(\boldsymbol{x}_i, \boldsymbol{y}_i) \overset{i.i.d.}{\sim} \mathcal{D}$ for $i = 1, \ldots, n$ is distributed according to some probability measure $\mathcal{D}$. We refer to $\boldsymbol{x}_i \in \mathcal{X} \subset \mathbb{R}^d$ as the input and to $\boldsymbol{y} \in \mathcal{Y} \subset \mathbb{R}^C$ as the target. We will often use matrix notations to obtain more compact formulations, thus summarizing all inputs and targets as matrices $\boldsymbol{X} \in \mathbb{R}^{n \times d}$ and $\boldsymbol{Y} \in \mathbb{R}^{n \times C}$. We will mainly be concerned with the task of multiclass classification, where targets $\boldsymbol{y}_i$ are encoded as one-hot vectors. We consider a **function space** $\mathcal{F}$ and an associated **loss function** $L_f : \mathcal{X} \times \mathcal{Y} \to \mathbb{R}$ that measures how well a given function $f \in \mathcal{F}$ performs at predicting the targets. More concretely, we define the regularized empirical error as

$$\hat{L}_{\mathcal{S}}^{\lambda} : \mathcal{F} \to \mathbb{R}, \quad f \mapsto \frac{1}{n} \sum_{i=1}^{n} L_f(\boldsymbol{x}_i, \boldsymbol{y}_i) + \lambda \Omega(f),$$

where $\lambda > 0$ is a regularization parameter and $\Omega : \mathcal{F} \to \mathbb{R}$ is a functional. For any vector $\boldsymbol{v} \in \mathbb{R}^C$, let $v^* = \operatorname{argmax}_{k \leq C} v_k$. Using this notation, we let $a_f(\boldsymbol{x}, \boldsymbol{y}) := \mathbb{1}_{\{f^*(\boldsymbol{x}) = y^*\}}$. We then define the **empirical accuracy** $\hat{A}_{\mathcal{S}}$ as the average number of instances for which $f \in \mathcal{F}$ provides the correct prediction, i.e.

$$\hat{A}_{\mathcal{S}} : \mathcal{F} \to [0, 1], \quad f \mapsto \frac{1}{n} \sum_{i=1}^{n} a_f(\boldsymbol{x}_i, \boldsymbol{y}_i)$$

Finally, we introduce a **learning algorithm** $\mathcal{Q}_{\mathcal{F}} : (\mathcal{X} \times \mathcal{Y})^n \to \mathcal{F}$, that given a function class $\mathcal{F}$ and a training set $\mathcal{S} \in (\mathcal{X} \times \mathcal{Y})^n$, chooses a function $f \in \mathcal{F}$. We will exclusively be concerned with learning through empirical risk minimization, i.e.

$$\mathcal{Q}_{\mathcal{F}}^{\lambda}(\mathcal{S}) := \hat{f}_{\mathcal{S}}^{\lambda} := \operatorname{argmin}_{f \in \mathcal{F}} \hat{L}_{\mathcal{S}}^{\lambda}(f),$$

and use the shortcut $\hat{f}_{\mathcal{S}} := \hat{f}_{\mathcal{S}}^{\lambda=0}$. In practice however, the most important measures are given by the **generalization** loss and accuracy of the model, defined as

$$L : \mathcal{F} \to \mathbb{R}, \quad f \mapsto \mathbb{E}_{(\boldsymbol{x}, \boldsymbol{y}) \sim \mathcal{D}} [L_f(\boldsymbol{x}, \boldsymbol{y})], \quad A : \mathcal{F} \to [0, 1], \quad f \mapsto \mathbb{P}_{(\boldsymbol{x}, \boldsymbol{y}) \sim \mathcal{D}} (f^*(\boldsymbol{x}) = y^*).$$

A central goal in machine learning is to control the difference between the empirical error $\hat{L}_{\mathcal{S}}(f)$ and the generalization error $L(f)$. In this work, we mainly focus on the mean squared loss,

$$L_f(\boldsymbol{x}, \boldsymbol{y}) = \frac{1}{2} \sum_{k=1}^{C} (f_k(\boldsymbol{x}) - y_k)^2,$$

and also treat classification as a regression task, as commonly done in the literature (Lee et al., 2018; Chen et al., 2020; Arora et al., 2019a; Hui & Belkin, 2021; Bachmann et al., 2021). Finally, we consider the function spaces induced by kernels and neural networks, presented in the following.

### 3.1 KERNEL LEARNING

A kernel $K : \mathcal{X} \times \mathcal{X} \to \mathbb{R}$ is a symmetric, positive semi-definite function, i.e. $K(\boldsymbol{x}, \boldsymbol{x}') = K(\boldsymbol{x}', \boldsymbol{x}) \; \forall \, \boldsymbol{x}, \boldsymbol{x}' \in \mathcal{X}$ and for any set $\{\boldsymbol{x}_1, \ldots, \boldsymbol{x}_n\} \subset \mathcal{X}$, the matrix $\boldsymbol{K} \in \mathbb{R}^{n \times n}$ with entries $K_{ij} = K(\boldsymbol{x}_i, \boldsymbol{x}_j)$ is positive semi-definite. It can be shown that a kernel induces a *reproducing kernel Hilbert space*, a function space equipped with an inner product $\langle \cdot, \cdot \rangle_{\mathcal{H}}$, containing elements

$$\mathcal{H} = \operatorname{cl}\left(\left\{f : f_k(\cdot) = \sum_{i=1}^{n} \alpha_{ik} K(\cdot, \boldsymbol{x}_i) \text{ for } \boldsymbol{\alpha} \in \mathbb{R}^{n \times C}, \boldsymbol{x}_i \in \mathcal{X}\right\}\right)$$

where cl is the closure operator. Although $\mathcal{H}$ is an infinite dimensional space, it turns out that the minimizer $\hat{f}_{\mathcal{S}}^{\lambda}$ can be obtained in closed form for mean squared loss, given by

$$\hat{f}_{\mathcal{S}}^{\lambda}(\boldsymbol{x}) = \boldsymbol{K}_{\boldsymbol{x}}^T (\boldsymbol{K} + \lambda \mathbf{1}_n)^{-1} \boldsymbol{Y} \xrightarrow{\lambda \to 0} \hat{f}_{\mathcal{S}}(\boldsymbol{x}) := \boldsymbol{K}_{\boldsymbol{x}}^T \boldsymbol{K}^{\dagger} \boldsymbol{Y}$$

where the regularization functional is induced by the inner product of the space, $\Omega(f) = ||f||_{\mathcal{H}}^2$, $\boldsymbol{K} \in \mathbb{R}^{n \times n}$ has entries $K_{ij} = K(\boldsymbol{x}_i, \boldsymbol{x}_j)$ and $\boldsymbol{K_x} \in \mathbb{R}^n$ has entries $(K_{\boldsymbol{x}})_i = K(\boldsymbol{x}, \boldsymbol{x}_i)$. $\boldsymbol{A}^\dagger$ denotes the pseudo-inverse of $\boldsymbol{A} \in \mathbb{R}^{n \times n}$. We will analyze both the regularized ($\hat{f}_{\mathcal{S}}^\lambda$) and unregularized model ($\hat{f}_{\mathcal{S}}$) and their associated LOO errors in the following. For a more in-depth discussion of kernels, we refer the reader to Hofmann et al. (2008); Paulsen & Raghupathi (2016).

## 3.2 Neural Networks and Associated Kernels

Another function class that has seen a rise in popularity is given by the family of fully-connected neural networks of depth $L \in \mathbb{N}$,

$$\mathcal{F}_{NN} = \left\{ f_{\boldsymbol{\theta}} : f_{\boldsymbol{\theta}}(\boldsymbol{x}) = \boldsymbol{W}^{(L)} \sigma \left( \boldsymbol{W}^{(L-1)} \ldots \sigma \left( \boldsymbol{W}^{(1)} \boldsymbol{x} \right) \ldots \right), \text{ for } \boldsymbol{W}^{(l)} \in \mathbb{R}^{d_l \times d_{l-1}} \right\},$$

for $d_l \in \mathbb{N}$, $d_0 = d$, $d_L = C$ and $\sigma : \mathbb{R} \to \mathbb{R}$ is an element-wise non-linearity. We denote by $\boldsymbol{\theta} \in \mathbb{R}^m$ the concatenation of all parameters $(\boldsymbol{W}^{(1)}, \ldots, \boldsymbol{W}^{(L)})$, where $m = \sum_{i=0}^{L-1} d_i d_{i+1}$ is the number of parameters of the model. The parameters are initialized randomly according to $W_{ij}^{(l)} \sim \mathcal{N}(0, \sigma_w^2)$ for $\sigma_w > 0$ and then adjusted to optimize the empirical error $\hat{L}_{\mathcal{S}}$, usually through an iterative procedure using a variant of gradient descent. Typically, the entire vector $\boldsymbol{\theta}$ is optimized via gradient descent, leading to a highly non-convex problem which does not admit a closed-form solution $\hat{f}$, in contrast to kernel learning. Next, we will review settings under which the output of a deep neural network can be approximated using a kernel formulation.

**Random Features.** Instead of training all the parameters $\boldsymbol{\theta}$, we restrict the function space by only optimizing over the top layer $\boldsymbol{W}^{(L)}$ and freezing the other variables $\boldsymbol{W}^{(1)}, \ldots, \boldsymbol{W}^{(L-1)}$ to their value at initialization. This restriction makes the model linear in its parameters and induces a finite-dimensional feature map

$$\phi_{\text{RF}} : \mathbb{R}^d \to \mathbb{R}^{d_{L-1}}, \quad \boldsymbol{x} \mapsto \sigma \left( \boldsymbol{W}^{(L-1)} \ldots \sigma \left( \boldsymbol{W}^{(1)} \boldsymbol{x} \right) \ldots \right).$$

Originally introduced by Rahimi & Recht (2008) to enable more efficient kernel computations, the random feature model has recently seen a strong spike in popularity and has been the topic of many theoretical works (Jacot et al., 2020; Mei & Montanari, 2021).

**Infinite Width.** Another, only recently established correspondence emerges in the infinite-width scenario. By choosing $\sigma_w = \frac{1}{\sqrt{d_l}}$ for each layer, Lee et al. (2018) showed that at initialization, the network exhibits a Gaussian process behaviour in the infinite-width limit. More precisely, it holds that $f_k \overset{i.i.d.}{\sim} \mathcal{GP}(0, \Sigma^{(L)})$, for $k = 1, \ldots, C$, where $\Sigma^{(L)} : \mathbb{R}^d \times \mathbb{R}^d \to \mathbb{R}$ is a kernel, coined NNGP, obeying the recursive relation

$$\Sigma^{(l)}(\boldsymbol{x}, \boldsymbol{x}') = \mathbb{E}_{\boldsymbol{z} \sim \mathcal{N}(\boldsymbol{0}, \boldsymbol{\Sigma}^{(l-1)}|_{\boldsymbol{x}, \boldsymbol{x}'})} \left[ \sigma(z_1) \sigma(z_2) \right]$$

with base $\Sigma^{(1)}(\boldsymbol{x}, \boldsymbol{x}') = \frac{1}{\sqrt{d}} \boldsymbol{x}^T \boldsymbol{x}'$ and $\boldsymbol{\Sigma}^{(l-1)}|_{\boldsymbol{x}, \boldsymbol{x}'} \in \mathbb{R}^{2 \times 2}$ is $\Sigma^{(l-1)}$ evaluated on the set $\{\boldsymbol{x}, \boldsymbol{x}'\}$. To reason about networks trained with gradient descent, Jacot et al. (2018) extended this framework and proved that the empirical neural tangent kernel converges to a constant kernel $\Theta$. The so-called neural tangent kernel $\Theta : \mathbb{R}^d \times \mathbb{R}^d \to \mathbb{R}$ is also available through a recursive relation involving the NNGP:

$$\Theta^{(l+1)}(\boldsymbol{x}, \boldsymbol{x}') = \Theta^{(l)}(\boldsymbol{x}, \boldsymbol{x}') \dot{\Sigma}^{(l+1)}(\boldsymbol{x}, \boldsymbol{x}') + \Sigma^{(l+1)}(\boldsymbol{x}, \boldsymbol{x}'),$$

with base $\Theta^{(1)}(\boldsymbol{x}, \boldsymbol{x}') = \Sigma^{(1)}(\boldsymbol{x}, \boldsymbol{x}')$ and $\dot{\Sigma}^{(l)}(\boldsymbol{x}, \boldsymbol{x}') = \mathbb{E}_{\boldsymbol{z} \sim \mathcal{N}(\boldsymbol{0}, \boldsymbol{\Sigma}^{(l-1)}|_{\boldsymbol{x}, \boldsymbol{x}'})} \left[ \dot{\sigma}(z_1) \dot{\sigma}(z_2) \right]$. Training infinitely-wide networks with gradient descent thus reduces to kernel learning with the NTK $\Theta^{(L)}$. Arora et al. (2019a); Lee et al. (2019) extended this result to a non-asymptotic setting, showing that very wide networks trained with small learning rates become indistinguishable from kernel learning.

## 3.3 Leave-one-out Error

The goal of learning is to choose a model $f \in \mathcal{F}$ with minimal generalization error $L(f)$. Since the data distribution $\mathcal{D}$ is usually not known, practitioners obtain an estimate for $L(f)$ by using an

additional test set $\mathcal{S}_{\text{test}} = \{(\boldsymbol{x}_1^t, \boldsymbol{y}_1^t), \ldots, (\boldsymbol{x}_b^t, \boldsymbol{y}_b^t)\}$ not used as input to the learning algorithm $\mathcal{Q}_{\mathcal{F}}$, and calculate

$$L_{\text{test}} = \frac{1}{b} \sum_{i=1}^{b} L_f(\boldsymbol{x}_i^t, \boldsymbol{y}_i^t), \qquad A_{\text{test}} = \frac{1}{b} \sum_{i=1}^{b} a_f(\boldsymbol{x}_i^t, \boldsymbol{y}_i^t) \tag{1}$$

While this is a simple approach, practitioners cannot always afford to put a reasonably sized test set aside. This observation led the field of statistics to develop the idea of a leave-one-out error (LOO), allowing one to obtain an estimate of $L(f)$ without having to rely on additional data (Lachenbruch, 1967; Mosteller & Tukey, 1968). We give a formal definition below.

**Definition 3.1.** *Consider a learning algorithm $\mathcal{Q}_{\mathcal{F}}$ and a training set $\mathcal{S} = \{(\boldsymbol{x}_i, \boldsymbol{y}_i)\}_{i=1}^{n}$. Let $\mathcal{S}_{-i} = \mathcal{S} \setminus \{(\boldsymbol{x}_i, \boldsymbol{y}_i)\}$ be the training set without the $i$-th data point $(\boldsymbol{x}_i, \boldsymbol{y}_i)$ and define $\hat{f}^{-i} := \mathcal{Q}_{\mathcal{F}}(\mathcal{S}_{-i})$, the corresponding model obtained by training on $\mathcal{S}_{-i}$. The leave-one-out loss/accuracy of $\mathcal{Q}_{\mathcal{F}}$ on $\mathcal{S}$ is then defined as the average loss/accuracy of $\hat{f}^{-i}$ incurred on the left out point $(\boldsymbol{x}_i, \boldsymbol{y}_i)$, given by*

$$L_{LOO}(\mathcal{Q}_{\mathcal{F}}; \mathcal{S}) = \frac{1}{n} \sum_{i=1}^{n} L_{\hat{f}^{-i}}(\boldsymbol{x}_i, \boldsymbol{y}_i), \qquad A_{LOO}(\mathcal{Q}_{\mathcal{F}}; \mathcal{S}) = \frac{1}{n} \sum_{i=1}^{n} a_{\hat{f}^{-i}}(\boldsymbol{x}_i, \boldsymbol{y}_i)$$

Notice that calculating $L_{\text{LOO}}$ can indeed be done solely based on the training set. On the other hand, using the training set to estimate $L(f)$ comes at a cost. Computing $\hat{f}^{-i}$ for $i = 1, \ldots, n$ requires fitting the given model class $n$ times. While this might be feasible for smaller models such as random forests, the method clearly reaches its limits for larger models and datasets used in deep learning. For instance, evaluating $L_{\text{LOO}}$ for ResNet-152 on *ImageNet* requires fitting a model with $6 \times 10^7$ parameters $10^7$ times, which clearly becomes extremely inefficient. At first glance, the concept of leave-one-out error therefore seems to be useless in the context of deep learning. However, we will now show that the recently established connection between deep networks and kernel learning gives us an efficient way to compute $L_{\text{LOO}}$. Another common technique is $K$-fold validation, where the training set is split into $K$ equally sized folds and the model is evaluated in a leave-one-out fashion on folds. While more efficient than the vanilla leave-one-out error, $K$-fold does not admit a closed-form solution, making the leave-one-out error more attractive in the setting we study.

In the following, we will use the shortcut $\hat{f}_{\mathcal{S}}^{\lambda} := \hat{f}^{\lambda}$. Surprisingly, it turns out that the space of kernels using $\mathcal{Q}_{\mathcal{F}}^{\lambda}$ with mean-squared loss admits a closed-form expression for the leave-one-out error, only relying on a *single model* obtained on the full training set $(\boldsymbol{X}, \boldsymbol{Y})$:

**Theorem 3.2.** *Consider a kernel $K : \mathbb{R}^d \times \mathbb{R}^d \to \mathbb{R}$ and the associated objective with regularization parameter $\lambda > 0$ under mean squared loss. Define $\boldsymbol{A} = \boldsymbol{K}(\boldsymbol{K} + \lambda \mathbf{1}_n)^{-1}$. Then it holds that*

$$L_{LOO}\left(\mathcal{Q}_{\mathcal{F}}^{\lambda}\right) = \frac{1}{n} \sum_{i=1}^{n} \sum_{k=1}^{C} \left(\Delta_{ik}^{\lambda}\right)^2, \qquad A_{LOO}\left(\mathcal{Q}_{\mathcal{F}}^{\lambda}\right) = \frac{1}{n} \sum_{i=1}^{n} \mathbb{1}_{\left\{\left(\boldsymbol{y}_i - \Delta_{i\bullet}^{\lambda}\right)^* = \boldsymbol{y}_i^*\right\}} \tag{2}$$

*where the residual $\Delta_{ik}^{\lambda} \in \mathbb{R}$ for $i = 1, \ldots, n$, $k = 1, \ldots, C$ is given by $\Delta_{ik}^{\lambda} = \frac{Y_{ik} - \hat{f}_k^{\lambda}(\boldsymbol{x}_i)}{1 - A_{ii}}$.*

For the remainder of this text, we will use the shortcut $L_{\text{LOO}}^{\lambda} := L_{\text{LOO}}\left(Q_{\mathcal{F}}^{\lambda}\right)$. While the expression for $L_{\text{LOO}}^{\lambda}$ for $C = 1$ has been known in the statistics community for decades (Stone, 1974) and more recently for general $C$ (Tacchetti et al., 2012; Pahikkala & Airola, 2016), the formula for $A_{\text{LOO}}^{\lambda}$ is to the best of our knowledge a novel result. We provide a proof of Theorem 3.2 in the Appendix A.1. Notice that Theorem 3.2 not only allows for an efficient computation of $L_{\text{LOO}}^{\lambda}$ and $A_{\text{LOO}}^{\lambda}$ but also provides us with a simple formula to assess the generalization behaviour of the model. It turns out that $L_{\text{LOO}}^{\lambda}$ captures a great variety of generalization phenomena, typically encountered in deep learning. Leveraging the neural tangent kernel together with Theorem 3.2 allows us to investigate their origin.

As a first step, we consider the zero regularization limit $\lambda \to 0$, i.e. $L_{\text{LOO}}^0$ since this is the standard setting considered in deep learning. A quick look at the formula reveals however that care has to be taken; as soon as the kernel is invertible, we have a $\frac{0}{0}$ situation.

**Corollary 3.3.** *Consider the eigendecomposition $\boldsymbol{K} = \boldsymbol{V} \operatorname{diag}(\boldsymbol{\omega})\boldsymbol{V}^T$ for $\boldsymbol{V} \in O(n)$ and $\boldsymbol{\omega} \in \mathbb{R}^n$. Denote its rank by $r = \operatorname{rank}(\boldsymbol{K})$. Then it holds that the residuals $\Delta_{ik}^\lambda \in \mathbb{R}$ can be expressed as*

$$\Delta_{ik}^\lambda(r) = \sum_{l=1}^n Y_{lk} \frac{\sum_{j=r+1}^n V_{ij}V_{lj} + \sum_{j=1}^r \frac{\lambda}{\lambda+\omega_j} V_{ij}V_{lj}}{\sum_{j=r+1}^n V_{ij}^2 + \sum_{j=1}^r \frac{\lambda}{\lambda+\omega_j} V_{ij}^2}$$

*Moreover in the zero regularization limit, i.e. $\lambda \to 0$, it holds that*

$$\Delta_{ik}^\lambda(r) \to \Delta_{ik}(r) = \begin{cases} \displaystyle\sum_{l=1}^n Y_{lk} \frac{\sum_{j=r+1}^n V_{ij}V_{lj}}{\sum_{j=r+1}^n V_{ij}^2} & \text{if } r < n \\ \displaystyle\sum_{l=1}^n Y_{lk} \frac{\sum_{j=1}^n \frac{1}{\omega_j} V_{ij}V_{lj}}{\sum_{j=1}^n \frac{1}{\omega_j} V_{ij}^2} & \text{if } r = n \end{cases}$$

**Remark.** At first glance for $\lambda = 0$, a distinct phase transition appears to take place when moving from $r = n - 1$ to $r = n$. Notice however that a small eigenvalue $\omega_n$ will allow for a smooth interpolation, i.e. $\Delta_{ik}(n) \xrightarrow{\omega_n \to 0} \Delta_{ik}(n-1)$ for $i = 1, \ldots, n, k = 1, \ldots, C$.

A similar result for the special case of linear regression and $r = n$ has been derived in Hastie et al. (2020). We now explore how well these formulas describe the complex behaviour encountered in deep models in the kernel regime.

# 4 LEAVE-ONE-OUT AS A GENERALIZATION PROXY

In this section, we study the derived formulas in the context of deep learning, through the lens of the neural tangent kernel and random feature models. Through empirical studies, we investigate LOO for varying sample sizes, different amount of noise in the targets as well as different regimes of overparametrization, in the case of infinitely-wide networks. We evaluate the models on the benchmark vision datasets *MNIST* (LeCun & Cortes, 2010) and *CIFAR10* (Krizhevsky & Hinton, 2009). To estimate the true generalization error, we employ the test sets as detailed in equation 1. We provide theoretical insights into double descent, establishing the explosion in loss around the interpolation threshold. Finally, we demonstrate the utility of LOO for state-of-the-art networks by predicting their transferability to new tasks. All experiments are performed in Jax (Bradbury et al., 2018) using the neural-tangents library (Novak et al., 2020).

## 4.1 FUNCTION OF SAMPLE SIZE

While known to exhibit only a small amount of bias, the LOO error may have a high variance for small sample sizes (Varoquaux, 2018). Motivated by the consistency results obtained for ridge regression in Patil et al. (2021), we study how the LOO error (and its variance) of an infinitely-wide network behaves as a function of the sample size $n$. In the following we thus interpret LOO as a function of $n$. In order to quantify how well LOO captures the generalization error for different sample regimes, we evaluate a 5-layer fully-connected NTK model for varying sample sizes on *MNIST* and *CIFAR10*. We display the resulting test and LOO losses in Figure 1. We plot the average result of 5 runs along with confidence intervals. We observe that both $L_{\text{LOO}}$ and $A_{\text{LOO}}$ closely follow their corresponding test quantity, even for very small sample sizes of around $n = 500$. As we increase the sample size, the variance of the estimator decreases and the quality of both LOO loss and accuracy improves even further, offering a very precise estimate at $n = 20000$.

## 4.2 RANDOM LABELS

Next we study the behaviour of the generalization error when a portion of the labels is randomized. In Zhang et al. (2017), the authors investigated how the performance of deep neural networks is affected when the functional relationship between inputs $\boldsymbol{x}$ and targets $\boldsymbol{y}$ is destroyed by replacing $\boldsymbol{y}$ by a random label. More specifically, given a dataset $\boldsymbol{X} \in \mathbb{R}^{n \times d}$ and one-hot targets $\boldsymbol{y} = \boldsymbol{e}_k$ for $k \in \{1, \ldots, C\}$, we fix a noise level $p \in [0, 1]$ and replace a subset of size $pn$ of the targets with

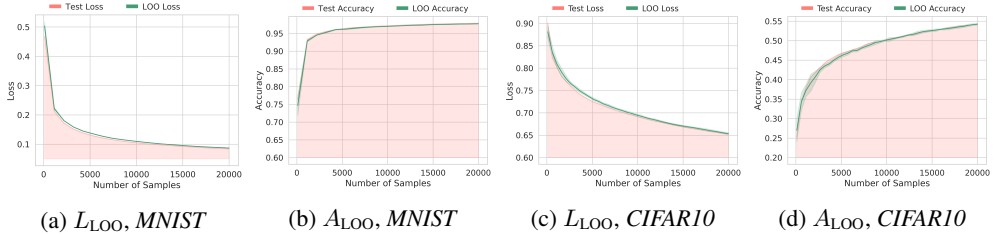

(a) $L_{\text{LOO}}$, *MNIST*  (b) $A_{\text{LOO}}$, *MNIST*  (c) $L_{\text{LOO}}$, *CIFAR10*  (d) $A_{\text{LOO}}$, *CIFAR10*

Figure 1: Test and LOO losses (a,c) and accuracies (b, d) as a function of sample size $n$. We use a 5-layer fully-connected NTK model on *MNIST* and *CIFAR10*.

random ones, i.e. $\boldsymbol{y} = \boldsymbol{e}_{\tilde{k}}$ for $\tilde{k} \sim \mathcal{U}(\{1, \ldots, C\})$. We then train the model using the noisy targets $\tilde{\boldsymbol{y}}$ while measuring the generalization error with respect to the clean target distribution. As observed in Zhang et al. (2017); Rolnick et al. (2018), neural networks manage to achieve very small empirical error even for $p = 1$ whereas their generalization consistently worsens with increasing $p$. Random labels has become a very popular experiment to assess the quality of generalization bounds (Arora et al., 2019b; 2018; Bartlett et al., 2017). We now show how the leave-one-out error exhibits the same behaviour as the test error under label noise, thus serving as an ideal test bed to analyze this phenomenon. In the following, we consider the leave-one-out error as a function of the labels, i.e. $L_{\text{LOO}} = L_{\text{LOO}}(\boldsymbol{y})$. The phenomenon of noisy labels is of particular interest when the model has enough capacity to achieve zero empirical error. For kernels, this is equivalent to $\text{rank}(\boldsymbol{K}) = n$ and we will thus assume in this section that the kernel matrix has full rank. However, a direct application of Theorem 3.2 is not appropriate, since randomizing some of the training labels also randomizes the "implicit" test set. In other words, our aim is not to evaluate $\hat{f}_{\tilde{\mathcal{S}}}^{-i}$ against $\tilde{\boldsymbol{y}}_i$ since $\tilde{\boldsymbol{y}}_i$ might be one of the randomized labels. Instead, we want to compare $\hat{f}_{\tilde{\mathcal{S}}}^{-i}$ with the true label $\boldsymbol{y}_i$, in order to preserve a clean target distribution. To fix this issue, we derive a formula for the leave-one-out error for a model trained on labels $\tilde{\boldsymbol{Y}}$ but evaluated on $\boldsymbol{Y}$:

**Proposition 4.1.** *Consider a kernel with spectral decomposition* $\boldsymbol{K} = \boldsymbol{V} \operatorname{diag}(\boldsymbol{\omega}) \boldsymbol{V}^T$ *for* $\boldsymbol{V} \in \mathbb{R}^{n \times n}$ *orthogonal and* $\boldsymbol{\omega} \in \mathbb{R}^n$. *Assume that* $\text{rank}(\boldsymbol{K}) = n$. *Then it holds that the leave-one-out error* $L_{LOO}(\tilde{\boldsymbol{Y}}; \boldsymbol{Y})$ *for a model trained on* $\tilde{\mathcal{S}} = \{(\boldsymbol{x}_i, \tilde{\boldsymbol{y}}_i)\}_{i=1}^n$ *but evaluated on* $\mathcal{S} = \{(\boldsymbol{x}_i, \boldsymbol{y}_i)\}_{i=1}^n$ *is given by*

$$L_{LOO}(\tilde{\boldsymbol{Y}}; \boldsymbol{Y}) = \frac{1}{n} \sum_{i=1}^n \sum_{k=1}^K \left( \tilde{\Delta}_{ik} + Y_{ik} - \tilde{Y}_{ik} \right)^2, \quad A_{LOO}(\tilde{\boldsymbol{Y}}; \boldsymbol{Y}) = \frac{1}{n} \sum_{i=1}^n \mathbb{1}_{\left\{ (\tilde{\boldsymbol{y}}_i - \tilde{\boldsymbol{\Delta}}_{i\bullet})^* = \boldsymbol{y}_i^* \right\}}$$

*where* $\tilde{\Delta}_{ik} = \Delta_{ik}(\tilde{\boldsymbol{Y}}) \in \mathbb{R}$ *is defined as in Corollary 3.3.*

We defer the proof to the Appendix A.4. Attentive readers will notice that we indeed recover Theorem 3.2 when $\tilde{\boldsymbol{Y}} = \boldsymbol{Y}$. Using this generalization of LOO, we can study the setup in Zhang et al. (2017) to test whether LOO indeed correctly reflects the behaviour of the generalization error. To this end, we perform a random label experiment on *MNIST* and *CIFAR10* with $n = 20000$ for a NTK model of depth 5, where we track test and LOO losses while increasingly perturbing the targets with noise. We display the results in Figure 2. We see an almost perfect match between test and LOO loss as well test and LOO accuracy, highlighting how precisely LOO reflects the true error.

## 4.3 DOUBLE DESCENT

Originally introduced by Belkin et al. (2019), the double descent curve describes the peculiar shape of the generalization error as a function of the model complexity. The error first follows the classical $U$-shape, where an increase in complexity is beneficial until a specific threshold. Then, increasing the model complexity induces overfitting, which leads to worse performance and eventually a strong spike around the interpolation threshold. However, as found in Belkin et al. (2019), further increasing the complexity again reduces the error, often leading to the overall best performance. A large body of works provide insights into the double descent curve but most of them rely on very

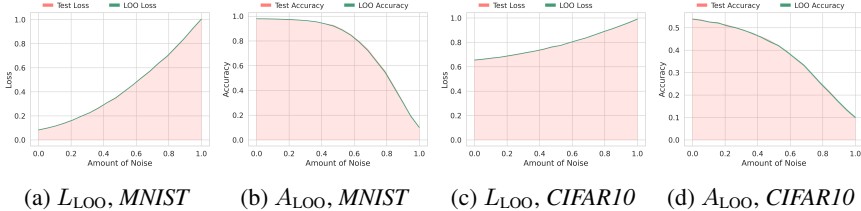

(a) $L_{\text{LOO}}$, *MNIST*  (b) $A_{\text{LOO}}$, *MNIST*  (c) $L_{\text{LOO}}$, *CIFAR10*  (d) $A_{\text{LOO}}$, *CIFAR10*

Figure 2: Test and LOO losses (a, c) and accuracies (b, d) as a function of noise. We use a 5-layer fully-connected NTK model on *MNIST* and *CIFAR10* with $n = 20000$.

restrictive assumptions such as Gaussian input distributions (Harzli et al., 2021; Adlam & Pennington, 2020; d'Ascoli et al., 2020) or teacher models (Bartlett et al., 2020; Harzli et al., 2021; Adlam & Pennington, 2020; Mei & Montanari, 2021; Hastie et al., 2020). Here we show under arguably weaker assumptions that the LOO loss also exhibits a spike at the interpolation threshold.

In order to reason about the effect of complexity, we have to consider models with a finite dimensional feature map $\phi_m : \mathbb{R}^d \to \mathbb{R}^m$. Increasing the dimensionality of the feature space $\mathbb{R}^m$ naturally overparametrizes the model and induces a dynamic in the rank $r(m) := \text{rank}(\boldsymbol{K}) = \text{rank}(\phi_m(\boldsymbol{X}))$. We restrict our attention to models that admit an interpolation point, i.e. $\exists m^* \in \mathbb{N}$ such that $r(m^*) = n$. While the developed theory holds for any kernel, we focus the empirical evaluations largely to the random feature model due to its aforementioned connection to neural networks. In the following, we will interpret the leave-one-out-error as a function of $m$ and $n$, i.e. $L_{\text{LOO}} = L_{\text{LOO}}^n(m)$. Notice that $L_{\text{LOO}}^n(m)$ measures the generalization error of a model trained on $n - 1$ points, instead of $n$. The underlying interpolation point thus shifts, i.e. we denote $m^* \in \mathbb{N}$ such that $r(m^*) = n - 1$. In the following we will show that $L_{\text{LOO}}^n(m^*(n)) \xrightarrow{n \to \infty} \infty$.

**Intuition.** We give a high level explanation for our theoretical insights in this paragraph. For $m < m^*$ the dominating term in $L_{\text{LOO}}^n(m)$ is

$$g(r, \lambda) = \sum_{i=1}^{n} \frac{1}{\sum_{l=r+1}^{n} V_{il}^2 + \sum_{l=1}^{r} \frac{\lambda}{\lambda + \omega_l} V_{il}^2}$$

We observe a blow-up for small $\lambda$ due to the unit vector nature of both $\boldsymbol{V}_{i\bullet}$ and $\boldsymbol{V}_{\bullet j}$. For instance, considering the interpolation point where $r = n - 1$ and $\lambda = 0$, we get $g(n-1, 0) = \sum_{i=1}^{n} \frac{1}{V_{in}^2}$, the sum over the elements of $\boldsymbol{V}_{\bullet n}$, which likely has a small entry due to its unit norm nature. On the other hand, reducing $r$ or increasing $\lambda$ has a dampening effect as we combine multiple positive terms together, increasing the likelihood to move away from the singularity. A similar dampening behaviour in $L_{\text{LOO}}^n(m)$ can also be observed for $m > m^*$ when $\omega_n >> 0$. In order to formalize this argument, we need the following assumption:

**Assumption 4.2.** *For $\boldsymbol{K} = \boldsymbol{V} \operatorname{diag}(\boldsymbol{\omega}) \boldsymbol{V}^T$ we impose that the orthogonal matrix $\boldsymbol{V}$ follows*

$$\boldsymbol{V} \sim \mathcal{U}(O_n)$$

*where $\mathcal{U}(O_n)$ is the Haar measure on the orthogonal group $O_n$.*

Under Assumption 4.2, we can prove that at the interpolation point, the leave-one-out error exhibits a spike, which worsens as we increase the amount of data:

**Theorem 4.3.** *For large enough $n \in \mathbb{N}$ and $\lambda \to 0$, it holds that*

$$L_{LOO}^n(m^*) \gtrsim 2nA$$

*where $A \sim \Gamma(\frac{1}{2}, 1)$ is independent of $n$. $L_{LOO}^n(m^*)$ hence diverges a.s. with $n \to \infty$.*

It is surprising that Assumption 4.2 suffices to prove Theorem 4.3, highlighting the universal nature of double descent. We provide empirical evidence for the double descent behaviour of LOO in Figure 3. We fit a 1-layer random feature model of varying width to the binary (labeled) *MNIST* dataset with a sample size of $n = 5000$. We observe that the LOO error indeed closely follows the

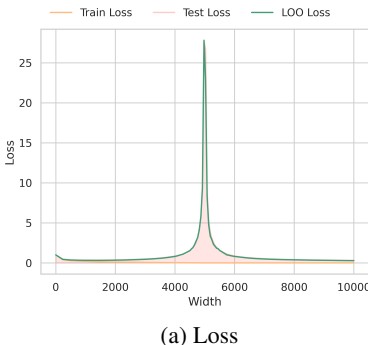 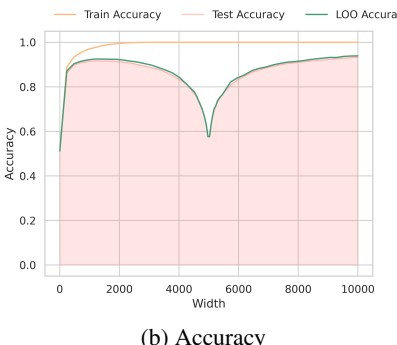

(a) Loss                                              (b) Accuracy

Figure 3: (a) Train, Test and LOO losses (a) and accuracies (b) as a function of width. We use a random feature model on binary *MNIST* with $n = 5000$.

test error, exhibiting the spike as predicted by Theorem 4.3. The location of the spike indeed exactly matches the interpolation threshold which is situated around $m^* \approx n$.

## 4.4 TRANSFER LEARNING

Finally, we study more realistic networks often employed for large-scale computer vision tasks. We study the task of transfer learning (Yosinski et al., 2014) by considering classifiers trained on *ImageNet* (Krizhevsky et al., 2012) and fine-tuning their top-layer to the *CIFAR10* dataset. Notice that this corresponds to training a kernel with a data-dependent feature map $\phi_{\text{data}}$ induced by the non-output layers. Our experiments on *ResNet18* (He et al., 2016), *AlexNet* (Krizhevsky et al., 2012), *VGG* (Simonyan & Zisserman, 2015) and *DenseNet* (Huang et al., 2017) demonstrate that also for transferring between tasks we have a very precise estimation of the test accuracy. We display the results in Table 1. We consider a small data regime where $n = 10000$, as often encountered in practice for transfer learning tasks. In order to highlight the role of pre-training, we also evaluate random feature maps $\phi_{\text{rand}}$ where we use standard initialization for all the parameters in the network. Indeed we clearly observe the benefits of pre-training, leading to a high increase in performance for all considered models. For both settings, pre-trained and randomly initialized, we observe an excellent agreement between test and leave-one-out accuracies across all architectures.

| MODEL | $A_{\text{TEST}}(\phi_{\text{DATA}})$ | $A_{\text{LOO}}(\phi_{\text{DATA}})$ | $A_{\text{TEST}}(\phi_{\text{RAND}})$ | $A_{\text{LOO}}(\phi_{\text{RAND}})$ |
|---|---|---|---|---|
| RESNET18 | $67.2 \pm 0.1$ | $67.5 \pm 0.4$ | $37.7 \pm 0.2$ | $37.1 \pm 0.1$ |
| ALEXNET | $57.6 \pm 0.2$ | $58.2 \pm 0.2$ | $24.2 \pm 0.2$ | $23.7 \pm 0.2$ |
| VGG16 | $56.6 \pm 0.2$ | $56.8 \pm 0.5$ | $29.3 \pm 0.5$ | $29.2 \pm 0.8$ |
| DENSENET161 | $72.5 \pm 0.2$ | $71.3 \pm 0.3$ | $51.7 \pm 0.3$ | $49.9 \pm 0.3$ |

Table 1: Test and LOO accuracies for models pre-trained on *ImageNet* and transferred to *CIFAR10* by re-training the top layer. We use 5 runs, each with a different training set of size $n = 10000$. We compare the pre-trained networks with random networks to show the benefits of transfer learning.

## 5 CONCLUSION

We investigated the concept of leave-one-out error as a measure for generalization in the context of deep learning. Through the use of NTK, we derive new expressions for the LOO loss and accuracy of deep models in the kernel regime and observe that they capture the generalization behaviour for a variety of settings. Moreover, we mathematically prove that LOO exhibits a spike at the interpolation threshold, needing only a minimal set of theoretical tools. Finally, we demonstrated that LOO accurately predicts the performance of deep models in the setting of transfer learning.

Notably, the simple form of LOO might open the door to new types of theoretical analyses to better understand generalization, allowing for a disentanglement of the roles of various factors at play such as overparametrization, noise and architectural design.

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

## A OMITTED PROOFS

We list all the omitted proofs in the following section.

### A.1 PROOF OF THEOREM 3.2

**Theorem 3.2.** *Consider a kernel $K : \mathbb{R}^d \times \mathbb{R}^d \to \mathbb{R}$ and the associated objective with regularization parameter $\lambda > 0$ under mean squared loss. Define $\boldsymbol{A} = \boldsymbol{K}\left(\boldsymbol{K} + \lambda \mathbf{1}_n\right)^{-1}$. Then it holds that*

$$
L_{LOO}\left(\mathcal{Q}_\lambda^{ERM}\right) = \frac{1}{n}\sum_{i=1}^{n}\sum_{k=1}^{K}\left(\Delta_{ik}^\lambda\right)^2, \quad A_{LOO}\left(\mathcal{Q}_\lambda^{ERM}\right) = \frac{1}{n}\sum_{i=1}^{n}\mathbb{1}_{\left\{(\boldsymbol{y}_i-\Delta_{i\bullet})^*=\boldsymbol{y}_i^*\right\}}
$$

*where the residuals $\Delta_{ik}^\lambda \in \mathbb{R}$ for $i = 1,\dots,n$, $k = 1,\dots,K$ is given by $\Delta_{ik}^\lambda = \frac{Y_{ik}-\hat{f}_k^\lambda(\boldsymbol{x}_i)}{1-A_{ii}}$.*

*Proof.* Recall that $\hat{f}_{\mathcal{S}}^\lambda$ solves the optimization problem

$$
\hat{f}_{\mathcal{S}}^\lambda = \operatorname{argmin}_{f\in\mathcal{F}} L_{\mathcal{S}}^\lambda(f) := \operatorname{argmin}_{f\in\mathcal{F}}\left\{\sum_{i=1}^{n}\sum_{k=1}^{K}\left(f_k(\boldsymbol{x}_i) - Y_{ik}\right)^2 + \lambda\|f\|_{\mathcal{H}}^2\right\}
$$

and predicting on the training data takes the form $\hat{f}_{\mathcal{S}}(\boldsymbol{X}) = \boldsymbol{A}\boldsymbol{Y}$ for some $\boldsymbol{A} \in \mathbb{R}^{n\times n}$. Now consider the model $f_\lambda^{-i} := \hat{f}_{\mathcal{S}_{-i}}^\lambda$ obtained from training on $\mathcal{S}_{-i}$. W.L.O.G. assume that $i = n$. We want to understand the quantity $\hat{f}_{\lambda,k}^{-n}(\boldsymbol{x}_n)$, i.e. the $k$-th component of the prediction on $\boldsymbol{x}_i$. To that end, consider the dataset $\mathcal{Z} := \mathcal{S}_{-n} \cup \left\{\left(\boldsymbol{x}_n, \hat{f}_\lambda^{-n}(\boldsymbol{x}_n)\right)\right\}$. Notice that for any $f \in \mathcal{F}$, it holds

$$
\begin{aligned}
L_{\mathcal{Z}}^\lambda(\hat{f}_\lambda^{-n}) &= \sum_{k=1}^{C}\left\{\sum_{i=1}^{n-1}\left(\hat{f}_{\lambda,k}^{-n}(\boldsymbol{x}_i) - Y_{ik}\right)^2 + \left(\hat{f}_{\lambda,k}^{-n}(\boldsymbol{x}_n) - \hat{f}_{\lambda,k}^{-n}(\boldsymbol{x}_n)\right)^2\right\} + \lambda\|\hat{f}_\lambda^{-n}\|_{\mathcal{H}}^2 \\
&= \sum_{k=1}^{C}\sum_{i=1}^{n-1}\left(\hat{f}_{\lambda,k}^{-n}(\boldsymbol{x}_i) - Y_{ik}\right)^2 + \lambda\|\hat{f}_\lambda^{-n}\|_{\mathcal{H}}^2 \\
&= L_{\mathcal{S}_{-n}}^\lambda\left(\hat{f}_\lambda^{-n}\right) \\
&\leq L_{\mathcal{S}_{-n}}^\lambda(f) \\
&\leq L_{\mathcal{S}_{-n}}^\lambda(f) + \sum_{k=1}^{C}\left(f_k(\boldsymbol{x}_n) - \hat{f}_{\lambda,k}^{-n}(\boldsymbol{x}_n)\right)^2 \\
&= L_{\mathcal{Z}}^\lambda(f)
\end{aligned}
$$

where the first inequality follows because $\hat{f}^{-n}$ minimizes $L_{\mathcal{S}_{-n}}^\lambda$ by definition. Thus $\hat{f}_\lambda^{-n}$ also minimizes $L_{\mathcal{Z}}^\lambda$ and hence also takes the form

$$
\hat{f}_\lambda^{-n}(\boldsymbol{X}) = \boldsymbol{A}\tilde{\boldsymbol{Y}}
$$

where $\tilde{\boldsymbol{Y}} \in \mathbb{R}^{n\times C}$ such that $\tilde{Y}_{ik} = \begin{cases} Y_{ik} & \text{if } i \neq n \\ \hat{f}_{\lambda,k}^{-n}(\boldsymbol{x}_n) & \text{else} \end{cases}$. Now we care about the $n$-th prediction which is

$$
\begin{aligned}
\hat{f}_{\lambda,k}^{-n}(\boldsymbol{x}_n) &= \sum_{j=1}^{n}A_{nj}\tilde{Y}_{jk} = \sum_{j=1}^{n-1}A_{nj}Y_{jk} + A_{nn}\hat{f}_{\lambda,k}^{-n}(\boldsymbol{x}_n) \\
&= \sum_{j=1}^{n}A_{nj}Y_{jk} - A_{nn}Y_{nk} + A_{nn}\hat{f}_{\lambda,k}^{-n}(\boldsymbol{x}_n) \\
&= \hat{f}_{\mathcal{S},k}^\lambda(\boldsymbol{x}_n) - A_{nn}Y_{nk} + A_{nn}\hat{f}_{\lambda,k}^{-n}(\boldsymbol{x}_n)
\end{aligned}
$$

Solving for $\hat{f}_{\lambda,k}^{-n}(\boldsymbol{x}_n)$ gives

$$\hat{f}_{\lambda,k}^{-n}(\boldsymbol{x}_n) = \frac{\hat{f}_{\mathcal{S},k}^{\lambda}(\boldsymbol{x}_n) - A_{nn}Y_{nk}}{1 - A_{nn}} \tag{3}$$

Then, subtracting $Y_{nk}$ leads to

$$Y_{nk} - \hat{f}_{\lambda,k}^{-n}(\boldsymbol{x}_n) = Y_{nk} - \frac{\hat{f}_{\mathcal{S},k}^{\lambda}(\boldsymbol{x}_n) - A_{nn}Y_{nk}}{1 - A_{nn}} = \frac{Y_{nk} - Y_{nk}A_{nn} - \hat{f}_{\mathcal{S},k}^{\lambda}(\boldsymbol{x}_n) + A_{nn}Y_{nk}}{1 - A_{nn}} = \frac{Y_{nk} - \hat{f}_{\mathcal{S},k}^{\lambda}(\boldsymbol{x}_n)}{1 - A_{nn}}$$

Squaring and summing the expression over $n$ and $k$ results in the formula.

For the accuracy, we know that we correctly predict if the maximal coordinate of $\hat{f}_{\lambda}^{-n}(\boldsymbol{x}_n)$ agrees with the maximal coordinate of $\boldsymbol{y}_n$, i.e.

$$\operatorname{argmax}_k \hat{f}_{\lambda,k}^{-n}(\boldsymbol{x}_n) = \operatorname{argmax}_k Y_{nk}$$

From equation 3, we notice that

$$\operatorname{argmax}_k \hat{f}_{\lambda,k}^{-n}(\boldsymbol{x}_n) = \operatorname{argmax}_k \frac{\hat{f}_{\mathcal{S},k}^{\lambda}(\boldsymbol{x}_n) - A_{nn}Y_{nk}}{1 - A_{nn}} = \operatorname{argmax}_k \frac{\hat{f}_{\mathcal{S},k}^{\lambda}(\boldsymbol{x}_n) - Y_{nk} + Y_{nk} - A_{nn}Y_{nk}}{1 - A_{nn}}$$

$$= \operatorname{argmax}_k -\Delta_{nk}^{\lambda} + Y_{nk}$$

$$= (\boldsymbol{y}_n - \boldsymbol{\Delta}_{n\bullet})^*$$

We thus have to check the indicator $\mathbb{1}_{\{(\boldsymbol{y}_n - \boldsymbol{\Delta}_{n\bullet})^* = \boldsymbol{y}_n^*\}}$ and sum it over $n$ to obtain the result. $\qquad\square$

## A.2 BINARY CLASSIFICATION

Here we state the corresponding results in the case of binary classification. The formulation for the accuracy changes slightly as now the sign of the classifier serves as the prediction.

**Proposition A.1.** *Consider a kernel $K : \mathbb{R}^d \times \mathbb{R}^d \to \mathbb{R}$ and the associated objective with regularization parameter $\lambda > 0$ under mean squared loss. Define $\boldsymbol{A} = \boldsymbol{K}(\boldsymbol{K} + \lambda \boldsymbol{1}_n)^{-1}$. Then it holds that*

$$L_{LOO}\left(\mathcal{Q}_{\lambda}^{ERM}\right) = \frac{1}{n}\sum_{i=1}^n \left(\Delta_i^{\lambda}\right)^2, \qquad A_{LOO}\left(\mathcal{Q}_{\lambda}^{ERM}\right) = \frac{1}{n}\sum_{i=1}^n \mathbb{1}_{\{y_i \Delta_i^{\lambda} < 1\}}$$

*where the residuals $\Delta_i^{\lambda} \in \mathbb{R}$ for $i = 1, \ldots, n$ is given by $\Delta_i^{\lambda} = \frac{y_i - \hat{f}^{\lambda}(\boldsymbol{x}_i)}{1 - A_{ii}}$.*

*Proof.* Notice that the result for $L_{\text{LOO}}$ is analogous to the proof for Theorem 3.2 by setting $K = 1$. For binary classification we use the sign of the classifier as a decision rule, i.e. the classifier predicts correctly if $y\hat{f}^{\lambda}(\boldsymbol{x}) > 0$. We can thus calculate that

$$y_n\hat{f}_{\lambda}^{-n}(\boldsymbol{x}_n) = \frac{y_n\hat{f}_{\mathcal{S}}^{\lambda}(\boldsymbol{x}_n) - A_{nn}y_n^2}{1 - A_{nn}} = \frac{y_n\hat{f}_{\mathcal{S}}^{\lambda}(\boldsymbol{x}_n) - y_n^2 + y_n^2 - y_n^2 A_{nn}}{1 - A_{nn}}$$

$$= y_n^2 - y_n\frac{y_n - \hat{f}_{\mathcal{S}}^{\lambda}(\boldsymbol{x}_n)}{1 - A_{nn}}$$

$$= 1 - y_n\frac{y_n - \hat{f}_{\mathcal{S}}^{\lambda}(\boldsymbol{x}_n)}{1 - A_{nn}}$$

$$= 1 - y_n\Delta_n^{\lambda}$$

Thus, the $n$-th sample is correctly classified if and only if

$$1 - y_n\Delta_n^{\lambda} > 0 \iff y_n\Delta_n^{\lambda} < 1$$

We now just count the correct predictions for the accuracy, i.e.

$$A_{\text{LOO}}\left(\mathcal{Q}_{\lambda}^{\text{ERM}}\right) = \frac{1}{n}\sum_{i=1}^n \mathbb{1}_{\{y_i \Delta_i^{\lambda} < 1\}}$$

$\square$

### A.3 PROOF OF COROLLARY 3.3

**Corollary 3.3.** *Consider the eigendecomposition $\boldsymbol{K} = \boldsymbol{V}\operatorname{diag}(\boldsymbol{\omega})\boldsymbol{V}^T$ for $\boldsymbol{V} \in O(n)$ and $\boldsymbol{\omega} \in \mathbb{R}^n$. Denote its rank by $r = \operatorname{rank}(\boldsymbol{K})$. Then it holds that the residuals $\Delta_{ik}^\lambda \in \mathbb{R}$ can be expressed as*

$$\Delta_{ik}^\lambda(r) = \sum_{l=1}^n Y_{lk} \frac{\sum_{k=r+1}^n V_{ik}V_{lk} + \sum_{k=1}^r \frac{\lambda}{\lambda+\omega_k} V_{ik}V_{lk}}{\sum_{k=r+1}^n V_{ik}^2 + \sum_{k=1}^r \frac{\lambda}{\lambda+\omega_k} V_{ik}^2}$$

*Moreover for zero regularization, i.e. $\lambda \to 0$, it holds that*

$$\Delta_{ik}^\lambda(r) \to \Delta_{ik}(r) = \begin{cases} \displaystyle\sum_{l=1}^n Y_{lk} \frac{\sum_{j=r+1}^n V_{ij}V_{lj}}{\sum_{j=r+1}^n V_{ij}^2} & \text{if } r < n \\ \displaystyle\sum_{l=1}^n Y_{lk} \frac{\sum_{j=1}^n \frac{1}{\omega_j} V_{ij}V_{lj}}{\sum_{j=1}^n \frac{1}{\omega_j} V_{ij}^2} & \text{if } r = n \end{cases}$$

*Proof.* Define $\boldsymbol{A} = \boldsymbol{K}(\boldsymbol{K} + \lambda\mathbf{1}_n)^{-1} \in \mathbb{R}^{n \times n}$. Recall that $\hat{f}_{\mathcal{S}}^\lambda(X) = \boldsymbol{A}\boldsymbol{y}$ and thus $\hat{f}_k^\lambda(\boldsymbol{x}_i) = \sum_{j=1}^n A_{ij}Y_{jk}$. Let us first simplify $\boldsymbol{A}$:

$$\boldsymbol{A} = \boldsymbol{K}(\boldsymbol{K} + \lambda\mathbf{1}_n)^{-1} = \boldsymbol{V}\operatorname{diag}(\boldsymbol{\omega})\boldsymbol{V}^T \left( \boldsymbol{V}\operatorname{diag}(\boldsymbol{\omega})\boldsymbol{V}^T + \lambda\mathbf{1}_n \right)^{-1}$$

$$= \boldsymbol{V}\operatorname{diag}\left( \frac{\boldsymbol{\omega}}{\boldsymbol{\omega} + \lambda} \right) \boldsymbol{V}^T$$

We can characterize the off-diagonal elements for $i \neq j$ as follows:

$$A_{ij} = \sum_{k=1}^n V_{ik} \frac{\omega_i}{\omega_i + \lambda} V_{jk} = \sum_{k=1}^r V_{ik}V_{jk} \frac{\omega_i}{\omega_i + \lambda} = \sum_{k=1}^r V_{ik}V_{jk} - \sum_{k=1}^r \frac{\lambda}{\omega_k + \lambda} V_{ik}V_{jk}$$

$$= -\sum_{k=r+1}^n V_{ik}V_{jk} - \sum_{k=1}^r \frac{\lambda}{\omega_k + \lambda} V_{ik}V_{jk}$$

where we made use of the fact that $\boldsymbol{V}_{i\bullet} \perp \boldsymbol{V}_{j\bullet}$. The diagonal elements $i = j$ on the other hand can be written as

$$A_{ii} = \sum_{k=1}^n V_{ik}^2 \frac{\omega_i}{\omega_i + \lambda} = \sum_{k=1}^r V_{ik}^2 \frac{\omega_k}{\omega_k + \lambda} = \sum_{k=1}^r V_{ik}^2 - \sum_{k=1}^r V_{ik}^2 \frac{\lambda}{\omega_k + \lambda}$$

$$= 1 - \sum_{k=r+1}^n V_{ik}^2 - \sum_{k=1}^r V_{ik}^2 \frac{\lambda}{\omega_k + \lambda}$$

where we have made use of the fact that $\boldsymbol{V}_{i\bullet}$ is a unit vector. Plugging-in the quantities into $L_{\text{LOO}}$ results in

$$\Delta_{ik}^\lambda = \frac{Y_{ik} - \hat{f}_k^\lambda(\boldsymbol{x}_i)}{1 - A_{ii}} = \frac{Y_{ik} - \sum_{l=1}^n A_{il}Y_{lk}}{1 - A_{ii}} = \frac{Y_{ik} - A_{ii}y_i - \sum_{l\neq i}^n A_{il}Y_{lk}}{1 - A_{ii}}$$

$$= \frac{Y_{ik}\left( \sum_{j=r+1}^n V_{ij}^2 + \sum_{k=1}^r V_{ij}^2 \frac{\lambda}{\omega_j + \lambda} \right) + \sum_{l\neq i}^n Y_{lk}\left( \sum_{j=r+1}^n V_{ij}V_{lj} + \sum_{j=1}^r \frac{\lambda}{\omega_j + \lambda} V_{ij}V_{lj} \right)}{\sum_{j=r+1}^n V_{ij}^2 + \sum_{j=1}^r V_{ij}^2 \frac{\lambda}{\omega_l + \lambda}}$$

$$= \frac{\sum_{l=1}^n Y_{lk}\left( \sum_{j=r+1}^n V_{ij}V_{lj} + \sum_{j=1}^r \frac{\lambda}{\omega_j + \lambda} V_{ij}V_{lj} \right)}{\sum_{j=r+1}^n V_{ij}^2 + \sum_{j=1}^r V_{ij}^2 \frac{\lambda}{\omega_k + \lambda}}$$

Now crucially, in the full rank case $r = n$, we have empty sums, i.e. $\sum_{l=r+1}^n = \sum_{l=n+1}^n = 0$ and we obtain

$$\Delta_{ik}^\lambda = \frac{\sum_{l=1}^n Y_{lk} \sum_{j=1}^n \frac{\lambda}{\omega_j + \lambda} V_{ij}V_{lj}}{\sum_{j=1}^n V_{ij}^2 \frac{\lambda}{\omega_j + \lambda}} = \frac{\sum_{l=1}^n Y_{lk} \sum_{j=1}^n \frac{1}{\omega_j + \lambda} V_{ij}V_{lj}}{\sum_{j=1}^n V_{ij}^2 \frac{1}{\omega_j + \lambda}}$$

$$\xrightarrow{\lambda \to 0} \sum_{l=1}^n Y_{lk} \frac{\sum_{j=1}^n \frac{1}{\omega_j} V_{ij}V_{lj}}{\sum_{j=1}^n \frac{1}{\omega_j} V_{ij}^2}$$

On the other hand, in the rank deficient case $r < n$ we can cancel the regularization term:

$$\Delta_{ik}^\lambda \xrightarrow{\lambda \to 0} \sum_{l=1}^n Y_{lk} \frac{\sum_{j=r+1}^n V_{ij}V_{lj}}{\sum_{j=r+1}^n V_{ij}^2}$$

Plugging this into the formulas for $L_{\text{LOO}}^\lambda$ and $A_{\text{LOO}}^\lambda$ concludes the proof. $\qquad\square$

## A.4    PROOF OF PROPOSITION 4.1

**Proposition 4.1.** *Consider a kernel with spectral decomposition $\boldsymbol{K} = \boldsymbol{V}\operatorname{diag}(\boldsymbol{\omega})\boldsymbol{V}^T$ for $\boldsymbol{V} \in \mathbb{R}^{n\times n}$ orthogonal and $\boldsymbol{\omega} \in \mathbb{R}^n$. Assume that $\operatorname{rank}(\boldsymbol{K}) = n$. Then it holds that the leave-one-out error $L_{LOO}(\tilde{\boldsymbol{Y}}; \boldsymbol{Y})$ for a model trained on $\tilde{\mathcal{S}} = \{(\boldsymbol{x}_i, \tilde{\boldsymbol{y}}_i)\}_{i=1}^n$ but evaluated on $\mathcal{S} = \{(\boldsymbol{x}_i, \boldsymbol{y}_i)\}_{i=1}^n$ is given by*

$$L_{LOO}(\tilde{\boldsymbol{Y}}; \boldsymbol{Y}) = \frac{1}{n}\sum_{i=1}^n\sum_{k=1}^C \left(\tilde{\Delta}_{ik} + Y_{ik} - \tilde{Y}_{ik}\right)^2, \qquad A_{LOO}(\tilde{\boldsymbol{Y}}; \boldsymbol{Y}) = \frac{1}{n}\sum_{i=1}^n \mathbb{1}_{\left\{(\tilde{\boldsymbol{y}}_i - \tilde{\boldsymbol{\Delta}}_{i\bullet})^* = \boldsymbol{y}_i^*\right\}}$$

*where $\tilde{\Delta}_{ik} = \Delta_{ik}(\tilde{\boldsymbol{Y}}) \in \mathbb{R}$ is defined as in Corollary 3.3.*

*Proof.* Denote by $\tilde{\mathcal{S}}_{-i}$ the dataset $\{(\boldsymbol{x}_j, \tilde{\boldsymbol{y}}_j)\}_{j\neq i}^n$. Denote by $\tilde{f}_\lambda^{-i}$ the model trained on $\tilde{\mathcal{S}}_{-i}$. Recall from the proof of Theorem 3.2 that

$$\tilde{f}_{\lambda,k}^{-n}(\boldsymbol{x}_n) = \frac{\tilde{f}_k^\lambda(\boldsymbol{x}_n) - A_{nn}\tilde{Y}_{nk}}{1 - A_{nn}}$$

Instead of subtracting the same label $\tilde{Y}_{nk}$, we now subtract the evaluation label $Y_{nk}$:

$$
\begin{aligned}
Y_{nk} - \tilde{f}_{\lambda,k}^{-n}(\boldsymbol{x}_n) &= \frac{Y_{nk} - A_{nn}Y_{nk} - \tilde{f}_k^\lambda(\boldsymbol{x}_n) + A_{nn}\tilde{Y}_{nk}}{1 - A_{nn}} = \frac{(1 - A_{nn})(Y_{nk} - \tilde{Y}_{nk}) + \tilde{Y}_{nk} - \tilde{f}_k^\lambda(\boldsymbol{x}_n)}{1 - A_{nn}} \\
&= (Y_{nk} - \tilde{Y}_{nk}) + \frac{\tilde{Y}_{nk} - \tilde{f}_k^\lambda(\boldsymbol{x}_n)}{1 - A_{nn}} \\
&= (Y_{nk} - \tilde{Y}_{nk}) + \tilde{\Delta}_{ik}^\lambda
\end{aligned}
$$

The second term $\tilde{\Delta}_{ik}^\lambda$ is now the summand of the standard leave-one-out error where we evaluate on $\tilde{\boldsymbol{Y}}$. We can hence re-use Theorem 3.3 to decompose it. Squaring and summing over $n$ concludes the LOO loss result. For the accuracy, we notice that a similar derivation as for Theorem 3.2 applies:

$$
\begin{aligned}
\operatorname{argmax}_k \tilde{f}_{\lambda,k}^{-n}(\boldsymbol{x}_n) &= \operatorname{argmax}_k \frac{\tilde{f}_k^\lambda(\boldsymbol{x}_n) - A_{nn}\tilde{Y}_{nk}}{1 - A_{nn}} = \operatorname{argmax}_k \frac{\hat{f}_k^\lambda(\boldsymbol{x}_n) - \tilde{Y}_{nk} + \tilde{Y}_{nk} - A_{nn}\tilde{Y}_{nk}}{1 - A_{nn}} \\
&= \operatorname{argmax}_k -\tilde{\Delta}_{nk}^\lambda + \tilde{Y}_{nk} \\
&= \left(\tilde{\boldsymbol{y}}_n - \tilde{\boldsymbol{\Delta}}_{n\bullet}\right)^*
\end{aligned}
$$

We thus have to check the indicator against the true label $\boldsymbol{y}_n$, i.e. $\mathbb{1}_{\left\{(\tilde{\boldsymbol{y}}_n - \tilde{\Delta}_{n\bullet})^* = \boldsymbol{y}_n^*\right\}}$ and sum it over $n$ to obtain the result. $\qquad\square$

## A.5    PROOF OF THEOREM 4.3

**Theorem 4.3.** *For large enough $n \in \mathbb{N}$, we can estimate as*

$$L_{LOO}^n(m^*) \gtrapprox 2nA$$

*where $A \sim \Gamma(\frac{1}{2}, 1)$ is independent of $n$. $L_{LOO}^n(m^*)$ hence diverges a.s. with $n \to \infty$.*

*Proof.* First we notice that for $m = m^*$, by definition it holds that $r = n - 1$, which simplifies the LOO expression to

$$L_{\text{LOO}}^n(m^*) \xrightarrow{\lambda \to 0} \frac{1}{n}\left(\sum_{i=1}^n \frac{1}{V_{in}^2}\right)\left(\sum_{i=1}^n y_i V_{in}\right)^2$$

For notational convenience, we will introduce $\boldsymbol{v} \in \mathbb{R}^n$ such that $v_i := V_{in}$. We will now bound the both factors one-by-one. The first part is a simple application of Proposition B.1 and Proposition B.5:

$$L_{LOO}^n(m^*) = \frac{1}{n}\left(\sum_{i=1}^n v_i\right)^2 \sum_{i=1}^n \frac{1}{v_i^2} \geq n^2 \frac{1}{n}\left(\sum_{i=1}^n v_i\right)^2 \stackrel{(d)}{=} n^2 B$$

Now for large enough $n$, we can use Lemma B.6 to make the following approximation in distribution:

$$nB \approx 2\frac{n-1}{2}B \xrightarrow{(d)} 2A$$

where $A \sim \Gamma(\frac{1}{2}, 1)$. Thus for large enough $n$, it holds that

$$L_{LOO}^n(m^*) \gtrsim 2nA$$

As the approximation becomes exact for larger and larger $n$, we conclude that

$$L_{LOO}^n(m^*) \xrightarrow{n \to \infty} \infty \quad \text{a.s.}$$

$\square$

# B    ADDITIONAL LEMMAS

In this section we present the additional technical Lemmas needed for the proofs of the main claims in A.

**Lemma B.1.** *Consider a unit vector $\boldsymbol{v} \in \mathbb{S}^{n-1}$. Then it holds that*

$$\sum_{i=1}^n \frac{1}{v_i^2} \geq n^2$$

*Proof.* Let's parametrize each $v_i$ as

$$v_i = \frac{z_i}{\sqrt{\sum_{i=1}^n z_i^2}}$$

for $i = 1, \ldots, n$ and $\boldsymbol{z} \in \mathbb{R}^n$. One can easily check that $||\boldsymbol{v}||_2 = 1$ and hence $\boldsymbol{v} \in \mathbb{S}^{n-1}$. Plugging this in, we arrive at

$$\sum_{i=1}^n \frac{1}{v_i^2} = \sum_{i=1}^n \frac{\sum_{j=1}^n z_j^2}{z_i^2} = \sum_{i=1}^n \sum_{j=1}^n \frac{z_j^2}{z_i^2} = n + \sum_{i=1}^n \sum_{j \neq i}^n \frac{z_j^2}{z_i^2}$$

We can re-arrange the sum into pairs

$$\frac{z_j^2}{z_i^2} + \frac{z_i^2}{z_j^2} = a^2 + \frac{1}{a^2} \geq 2$$

for $a^2 = \frac{z_j^2}{z_i^2} > 0$ and using the fact that $x + \frac{1}{x} \geq 2$ for $x \geq 0$. We can find $\frac{n(n-1)}{2}$ such summands, and thus

$$\sum_{i=1}^n \frac{1}{v_i^2} \geq n + 2\frac{n(n-1)}{2} = n^2$$

$\square$

**Lemma B.2.** *Consider $\boldsymbol{v} \sim \mathcal{U}\left(\mathbb{S}^{n-1}\right)$ and any fixed orthogonal matrix $\boldsymbol{U} \in O(n)$. Then it holds that*

$$\boldsymbol{U}\boldsymbol{v} \stackrel{(d)}{=} \boldsymbol{v}$$

*Proof.* This is a standard result and can for instance be found in Vershynin (2018). $\square$

**Lemma B.3.** *Consider $w \sim \mathcal{N}(\mathbf{0}, \mathbb{1}_n)$. Then it holds that*

$$v = \frac{w}{||w||_2} \sim \mathcal{U}\left(\mathbb{S}^{n-1}\right)$$

*Proof.* This is a standard result and can for instance be found in Vershynin (2018). $\square$

**Lemma B.4.** *Consider two independent Gamma variables $X \sim \text{Gamma}(\alpha, \nu)$ and $Y \sim \text{Gamma}(\beta, \nu)$. Then it holds that*

$$\frac{X}{X+Y} \sim \text{Beta}\left(\alpha, \beta\right)$$

*Proof.* This is a standard result and can for instance be found in Bowman et al. (1998). $\square$

**Lemma B.5.** *Consider $v \sim \mathcal{U}\left(\mathbb{S}^{n-1}\right)$. Then it holds that*

$$\frac{1}{n}\left(\sum_{i=1}^n y_i v_i\right)^2 \sim \text{Beta}\left(\frac{1}{2}, \frac{n-1}{2}\right)$$

*Proof.* First realize that we can write

$$\frac{1}{\sqrt{n}}\sum_{i=1}^n y_i v_i = \tilde{\mathbf{1}}_n^T\left(v \odot y\right) \stackrel{(d)}{=} \tilde{\mathbf{1}}_n^T v$$

where $\tilde{\mathbf{1}}_n = \left(\frac{1}{\sqrt{n}}, \ldots, \frac{1}{\sqrt{n}}\right)$ with $||\tilde{\mathbf{1}}_n||_2 = 1$ and the fact that $v \odot y \stackrel{(d)}{=} v$ for fixed $y \in \{-1, 1\}^n$. The idea is now to choose $U \in O(n)$ such that $U^T\tilde{\mathbf{1}}_n = e_1 = (1, 0, \ldots, 0)$. Then by using Lemma B.2, it holds

$$\tilde{\mathbf{1}}_n^T v \stackrel{(d)}{=} \tilde{\mathbf{1}}_n^T U v \stackrel{(d)}{=} \left(U\tilde{\mathbf{1}}_n\right)^T v \stackrel{(d)}{=} e_1^T v \stackrel{(d)}{=} v_1$$

Thus, surprisingly, it suffices to understand the distribution of $v_1$. By Lemma B.3, we know that

$$v_1 \stackrel{(d)}{=} \frac{z_1}{\sqrt{z_1^2 + \cdots + z_n^2}}$$

where $z \sim \mathcal{N}(\mathbf{0}, \mathbb{1}_n)$. We are interested in the square of this expression,

$$\frac{1}{n}\left(\sum_{i=1}^n v_i\right)^2 \stackrel{(d)}{=} v_1^2 \stackrel{(d)}{=} \frac{z_1^2}{z_1^2 + \cdots + z_n^2} \stackrel{(d)}{=} \frac{z_1^2}{z_1^2 + w}$$

where we define $w = \sum_{i=2}^n z_i^2$, clearly independent of $z_1^2$. Moreover, it holds that $z_1^2 \sim \text{Gamma}\left(\frac{1}{2}, \frac{1}{2}\right)$ and $w \sim \text{Gamma}\left(\frac{n-1}{2}, \frac{1}{2}\right)$. Thus, by Lemma B.4 we can conclude that

$$\frac{1}{n}\left(\sum_{i=1}^n v_i\right)^2 \sim \text{Beta}\left(\frac{1}{2}, \frac{n-1}{2}\right)$$

$\square$

**Lemma B.6.** *Consider the sequence of Beta distributions $X_n \sim \text{Beta}(k, n)$. Then it holds that*

$$nX_n \stackrel{(d)}{\longrightarrow} \text{Gamma}(k, 1)$$

*Proof.* This is a standard result and can for instance be found in Walck (1996). $\square$

## C FURTHER EXPERIMENTS

In this section we present additional experimental results on the leave-one-out error.

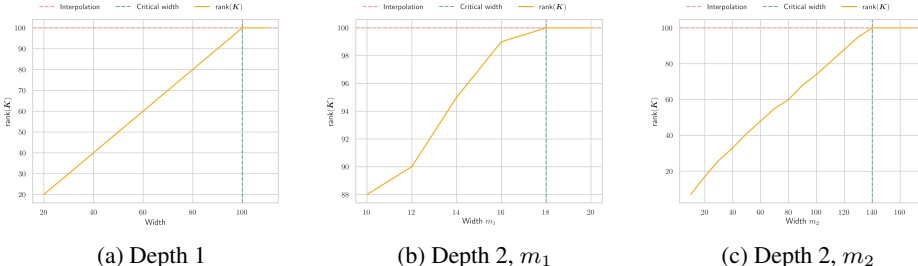

(a) Depth 1                (b) Depth 2, $m_1$                (c) Depth 2, $m_2$

Figure 4: Kernel rank $\mathrm{rank}(\boldsymbol{K})$ as a function of complexity. For (a) we use a depth 1 random feature model $\sigma(\boldsymbol{W}\boldsymbol{x})$, $\boldsymbol{W} \in \mathbb{R}^{m \times d}$, where complexity is measured through width $m$. In (b) and (c) we use a random feature model of depth 2, $\sigma(\boldsymbol{V}\sigma(\boldsymbol{W}\boldsymbol{x}))$, $\boldsymbol{W} \in \mathbb{R}^{m_1 \times d}$ and $\boldsymbol{V} \in \mathbb{R}^{m_2 \times m_1}$. For (b) we visualize the rank as a function of $m_1$, where $m_2 = 120$ fixed and in (c) as a function of $m_2$, where $m_1 = 10$ fixed. We use *MNIST* with $n = 100$ samples.

### C.1 Rank Dynamics for Different Kernels

In this section, we illustrate how the rank $\mathrm{rank}(\boldsymbol{K})$ of the kernel evolves as a function of the underlying complexity. As observed in various works, the spike in the double descent curve for neural networks usually occurs around the interpolation threshold, i.e. the point in complexity where the model is able to achieve zero training loss. Naturally, for the kernel formulation, interpolation is achieved when the kernel matrix $\boldsymbol{K}$ has full rank. We illustrate in the following how the dynamics of the rank change for different architectures, displaying a similar behaviour as finite width neural networks. In Fig. 4 (a) we show a depth 1 random feature model with feature maps $\sigma(\boldsymbol{W}\boldsymbol{x})$ where $\boldsymbol{W} \in \mathbb{R}^{m \times d}$ and $\sigma$ is the ReLU non-linearity. We use *MNIST* with $n = 100$. We measure the complexity through the width $m$ of the model. We observe that in this special case, the critical complexity exactly coincides with the number of samples, i.e. $\mathrm{rank}(\boldsymbol{K}) = n$ as soon as $m = n$. This seems counter-intuitive at first as an exact match of the complexity with the sample size is usually not observed for standard neural networks. We show however in Fig. 4 (b) and (c) that the rank dynamics indeed become more involved for larger depth. In Fig. 4 (b) and (c), we use a 2-layer random feature model with feature map $\sigma(\boldsymbol{V}\sigma(\boldsymbol{W}\boldsymbol{x}))$ for weights $\boldsymbol{W} \in \mathbb{R}^{m_1 \times d}$ and $\boldsymbol{V} \in \mathbb{R}^{m_2 \times m_1}$. In (b) we show the change in rank as $m_1$ increases, while $m_2 = 120$ is fixed. We observe that indeed the dynamics change and the critical complexity is not at $m_1 = n$. Similarly in (c) we fix $m_1 = 10$ and vary $m_2$. Also there we observe that the critical threshold is not located at $n$ but rather a bigger width is needed to reach interpolation.

Finally we also study the linearizations of networks that also give rise to kernels. More concretely, we consider feature maps of the form

$$\phi(\boldsymbol{x}) = \nabla_{\boldsymbol{\theta}} f_{\boldsymbol{\theta}}(\boldsymbol{x})$$

where $f_{\boldsymbol{\theta}}$ is a fully-connected network with parameters $\boldsymbol{\theta} \in \mathbb{R}^p$, as introduced before. Our double-descent framework also captures this scenario. In Fig. 5, we display the rank dynamics in case of a 2 layer network $f_{\boldsymbol{\theta}}(\boldsymbol{x}) = \boldsymbol{w}^T \sigma(\boldsymbol{U}\sigma(\boldsymbol{V}\boldsymbol{x}))$ where $\boldsymbol{w} \in \mathbb{R}^{m_2}$, $\boldsymbol{U} \in \mathbb{R}^{m_2 \times m_1}$, $\boldsymbol{V} \in \mathbb{R}^{m_1 \times d}$ and $\sigma$ is the ReLU non-linearity. Again we observe that the rank dynamics change and do not coincide with the number of samples $n$.

### C.2 LOO as function of depth $L$

We study how the depth $L$ of the NTK kernel $\Theta^{(L)}$ affects the performance of LOO loss and accuracy. We use the datasets *MNIST* and *CIFAR10* with $n = 5000$ and evaluate NTK models with depth ranging from 3 to 20. We present our findings in Figure 6. Again we see a very close match between LOO and the corresponding test quantity for *CIFAR10*. Interestingly the performance is slightly worse for very shallow models. For *MNIST* we see a gap between LOO loss and test loss, which is due to the very zoomed-in nature of the plot (the gap is actually only 0.015) as the loss values are very small in general. Indeed we observe an excellent match between the test and LOO accuracy.

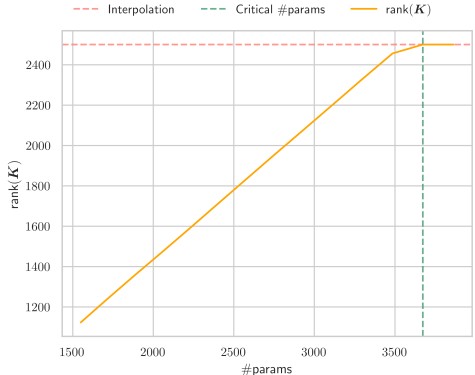

Figure 5: Rank dynamics of $K$ for the linearization kernel of $f_{\boldsymbol{\theta}}(\boldsymbol{x}) = \boldsymbol{w}^T \sigma(\boldsymbol{U}\sigma(\boldsymbol{V}\boldsymbol{x}))$ where $\boldsymbol{w} \in \mathbb{R}^{m_2}$, $\boldsymbol{U} \in \mathbb{R}^{m_2 \times m_1}$, $\boldsymbol{V} \in \mathbb{R}^{m_1 \times d}$ and $\sigma$ is the ReLU non-linearity.

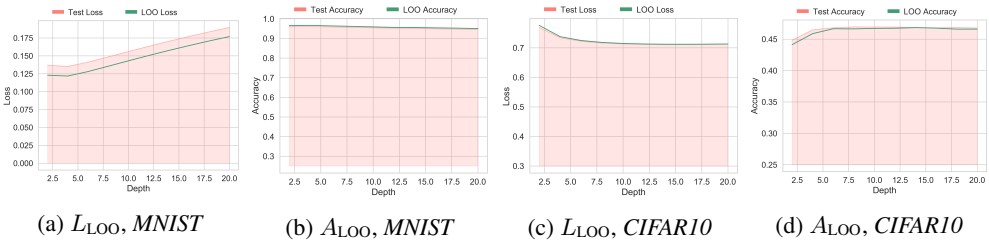

(a) $L_{\text{LOO}}$, *MNIST*    (b) $A_{\text{LOO}}$, *MNIST*    (c) $L_{\text{LOO}}$, *CIFAR10*    (d) $A_{\text{LOO}}$, *CIFAR10*

Figure 6: Test and LOO losses (a, c) and accuracies (b, d) as a function of depth $L$. We use fully-connected NTK model on *MNIST* and *CIFAR10*.

## C.3 Double Descent with Random Labels

Here we demonstrate how the spike in double descent is a very universal phenomenon as demonstrated by Theorem 4.3. We consider a random feature model of varying width $m$ on binary *MNIST* with $n = 2000$, where the labels are fully randomized ($p = 1$), destroying thus any relationship between the inputs and targets. Of course, there will be no double descent behaviour in the test accuracy as the network has to perform random guessing at any width. We display this in Figure 7. We observe that indeed the model is randomly guessing throughout all the regimes of overparametrization. Both the test and LOO loss however, exhibit a strong spike around the interpolation threshold. This underlines the universal nature of the phenomenon, connecting with the fact that Theorem 4.3 does not need any assumptions on the targets.

## C.4 Transfer Learning Loss

We report the corresponding test and leave-one-out losses, moved to the appendix due to space constraints. We display the scores in Table 2. Again we observe a very good match between the test and LOO losses. Moreover, we again find that pre-training on *ImageNet* is beneficial in terms of the achieved loss values.

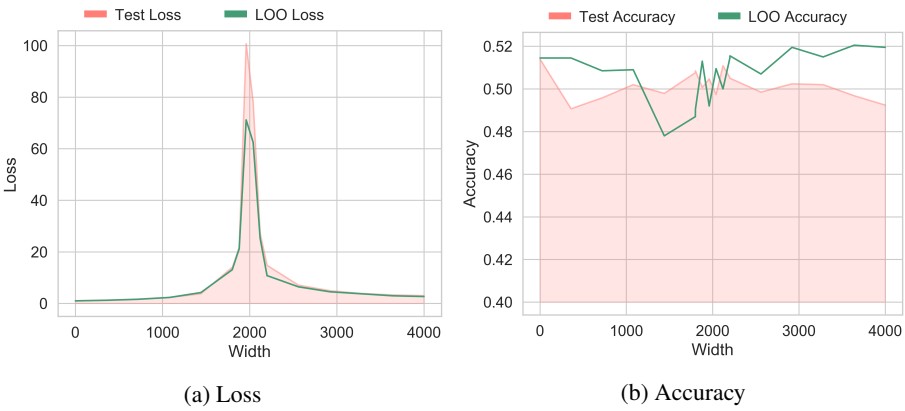

(a) Loss  (b) Accuracy

Figure 7: Test and LOO losses (a) and accuracies (b) as a function of sample width $m$. We use a random feature model on binary *MNIST* with random labels.

| MODEL | $L_{\text{TEST}}(\phi_{\text{DATA}})$ | $L_{\text{LOO}}(\phi_{\text{DATA}})$ | $L_{\text{TEST}}(\phi_{\text{RAND}})$ | $L_{\text{LOO}}(\phi_{\text{RAND}})$ |
|---|---|---|---|---|
| RESNET18 | $0.602 \pm 0.0053$ | $0.619 \pm 0.009$ | $0.77 \pm 0.003$ | $0.758 \pm 0.0051$ |
| ALEXNET | $0.638 \pm 0.0021$ | $0.639 \pm 0.0036$ | $0.837 \pm 0.002$ | $0.835 \pm 0.004$ |
| VGG16 | $0.657 \pm 0.0062$ | $0.66 \pm 0.0091$ | $0.852 \pm 0.0026$ | $0.834 \pm 0.0049$ |
| DENSENET161 | $0.599 \pm 0.0044$ | $0.613 \pm 0.0097$ | $0.718 \pm 0.0043$ | $0.731 \pm 0.0081$ |

Table 2: Test and LOO losses for models pre-trained on *ImageNet* and transferred to *CIFAR10* by re-training the top layer. We use 5 runs, each with a different training set of size $n = 10000$. We compare the pre-trained networks with random networks to illustrate the benefits of transfer learning.

