# OpenReview forum: "Generalization Through the Lens of Leave-One-Out Error"
_ICLR.cc/2022/Conference — ICLR 2022 Poster_

### Official Review · Reviewer_dZ2X · 2021-11-01

**Correctness:** 3
**Technical Novelty And Significance:** 3
**Empirical Novelty And Significance:** 3
**Recommendation:** 6
**Confidence:** 3

**Main Review:**

The idea to propose an alternative view on the generalization error estimation for deep learning is interesting and up to my knowledge novel. The proposed approach to avoid computational burden based on the closed form calculation of LOO error/accuracy makes sense when connected to the view of neural networks as NTK.

It is hard to judge the novelty of the closed form for LOO error/accuracy in multi-class setup, since (i) the loss considered is very exotic, being a sum of MSE of each element of a prediction vector compared to each element of the one-hot-encoded label (ii) there are multiple works deriving bounds on LOO for kernel machines (e.g., [1], [2], [3], [4]) none of which are mentioned in the related work. (i) leads to the fact that the central theorem of the paper follows from the cited work of M.Stone (1974) since there a general regression task is considered.

The Settings and Background section requires a lot of attentive reading out since the formal definitions are very sloppy. Some of the multiple points: {i} f* and y* are never defined (ii) accuracy is defined as "number of instances with correct prediction", but formalized to be in real numbers (iii) a training set S is defined as subset of all n-length sequences of X, when it is just one element of it - and should contain targets as well (iv) compact form of writing f is changing from reducing S, then to reducing \lambda. All these inconsistencies make it very hard to follow the formal flow of the paper. Analogously, the setup with neural networks does not specify that only fully-connected networks are considered.

My main concern though is the experimental setup. All of the experiments (except for one) and all the insights in generalization and double descent are produced for the kernel machines and not for neural networks. In my opinion this renders the claim of the paper contribution as generalization insights into deep learning models incorrect. While this still can be a valuable contribution to kernel optimization, in this current formulation I cannot agree with the contribution. In particular, for double descent experiment, the conclusion is that exactly the training size defines the spike in the complexity/error curve, which does not seem to be valid for neural networks.

Also I doubt the setup of the noise experiment, where LOO error peaks into the true label - this somehow makes the setup unfair since the generalization gap is not estimated from the training set only in such case.

Finally, I wonder why transfer learning experiments do not report the test loss approximation by LOO error, but includes the random initialization experiment, which is unrelated to the current work.


[1] Strobl, Eric V., and Shyam Visweswaran. "Deep multiple kernel learning." 2013 12th International Conference on Machine Learning and Applications. Vol. 1. IEEE, 2013.

[2] Dhar, Sauptik, Vladimir Cherkassky, and Mohak Shah. "Multiclass learning from contradictions." Advances in Neural Information Processing Systems 32 (2019): 8400-8410.

[3] Passerini, Andrea, Massimiliano Pontil, and Paolo Frasconi. "From Margins to Probabilities in Multiclass Learning." ECAI 2002: 15th European Conference on Artificial Intelligence, July 21-26, 2002, Lyon France: Including Prestigious Applications of Intelligent Systems (PAIS 2002): Proceedings. Vol. 77. IOS Press, 2002.

[4] Tuda, Koji, et al. "Learning to predict the leave-one-out error of kernel based classifiers." International Conference on Artificial Neural Networks. Springer, Berlin, Heidelberg, 2001.


**Summary Of The Paper:**

The paper proposes to approach the generalization prediction problem in the domain of big neural networks using the classical notion of leave-one-out (LOO) error. Such metric allows to find generalization error/accuracy using only the training dataset. The authors demonstrate, that following the results presented earlier, one can derive a closed form for LOO error/accuracy for multiclass classification tasks and kernel space, thus allowing not to run expensive calculations. Empirically, authors investigate NTK generalization properties, in the context of noisy labels and double descent investigation. They also demonstrate accuracy prediction for fine-tuning task on neural networks.

**Summary Of The Review:**

Based on the concerns with the experimental setup and what exactly it shows I recommend to review the perspective at which the results are presented. In the current form the paper cannot be accepted.

---
Based on the authors' explanations and editions of the paper I change my score.

---

> ### Author Response · Authors · 2021-11-19
> **Answer to Reviewer 3 (I)**
>
> Thank you for your feedback. We will address your questions and comments in the following.
>
> *Loss is very exotic:*
> Please note that the considered loss is a simple (multi-class) MSE loss, this is in no way an "exotic" formulation but rather a standard one. It emerges naturally as the limit of a neural network trained on (multi-class) MSE loss as the widths go to infinity, as shown in the seminal work of Jacot et al. The fact that the prediction vector is shared among different classes is due to the tangent kernel being the same for each class in the limit. This is thus not a weakness or an assumption but rather a consequence of the behaviour of very wide networks. On the other hand, considering classification as a regression task is a very common approach in theoretical works, see the references provided in the main text and the additional reference that Reviewer 2 pointed out.
>
> *Additional Citations:*
> Thank you for pointing out these additional works. We want to highlight however that we work with an exact formula for the accuracy, while these papers derive upper bounds. We will include these papers in our work and mention this difference.
>
> *LOO formulas are not novel but follow from Stone:*
> Thank you for raising this concern, we however respectfully disagree. While the initial framework considered by Stone is very general, the only closed-form solution he derives (relevant to this work) is in case of the mean squared loss with one class (for which we properly cite it). We could not find an extension to LOO accuracy nor the case of zero regularization, which is very central to our work.
>
> *Sloppy writing and inconsistencies:*
> We apologize that the Setting and Background section required attentive reading. We want to highlight however that other reviewers did not have the same concerns. The flaws you highlight are easy to fix in a revision, as explained next:
> i) The * notation is introduced exactly one sentence before f*  and y^* are used.
> ii) We properly define accuracy, and while we formalize it to take values in the real numbers instead of [0,1] (which is mathematically not incorrect), we think that this is very minor and should not hinder the readability of our work.
> iii) We agree that it is an element of the space (\mathcal{X} \times \mathcal{Y})^{n} and not a subset. The inaccuracy is however minor and we do properly define \mathcal{S} on page 2.
> iv) It is correct that we suppress different quantities to reduce clutter but we introduce each short cut properly, so there should not be any confusion.
>
> We have corrected the inconsistencies you point out to make the paper as accessible as possible.
>
>
> *Infinite vs finite width networks:*
> Thank you for raising this, you are correct, we estimate the generalization performance of the underlying infinitely-wide network which corresponds to learning with the neural tangent kernel. We did not perform experiments for finite width networks since there is a large body of theoretical work, proving how very wide networks become essentially indistinguishable from their kernel counterpart [Arora et al., 2019, Lee et al., 2019].  We apologize if our writing suggests that we deal with finite width networks out of the kernel regime and we will rephrase to make this point absolutely clear to the reader. We would however like to remark that understanding the generalization of such wide networks is still far from complete and LOO could serve as a great tool to advance this understanding. The LOO error is hence very relevant to deep learning and not just to kernel theory. Finally we also demonstrate the utility of LOO for state-of-the-art networks, where we can accurately predict the performance of a network on a new task.

---

> > ### Author Response · Authors · 2021-11-19
> > **Answer to Reviewer 3 (II)**
> >
> > *Spike coincides with number of samples*
> > We agree with your statement, indeed finite neural networks do not usually display the spike at the point where the complexity (for instance parameter count) reaches the sample size but rather at the interpolation point. In our case, the rank of the kernel matrix dictates at which complexity we reach the interpolation point, i.e. when it is full rank. In the experiment we used a 1 hidden layer random feature model for which it holds that the rank of the kernel matrix is exactly the width of the model. For this reason, it holds that the spike appears exactly when the width reaches the number of samples. In general however, this is usually not the case. We added a paragraph in the appendix (Section C.1) detailing this. We also display the rank as a function of complexity for different models (Fig. 4a), Fig. 4b), Fig.4c and Fig.5), showing that indeed the interpolation point shifts for different architectures and does not simply coincide with the number of samples.
> >
> >
> > *Noise experiment "peaks" into test data*
> > This is a great observation, indeed we assume that the true labels from the training set are known. We are however mainly interested in the phenomenon of "noisy labels" and not necessarily in practical applications. We completely agree that in a practical setting, where the training data has noisy examples (but we do not know which) and we are interested in the performance of the model on clean data, the LOO cannot help as it can only provide an estimate of the performance on noisy test data. We are however more interested in understanding the robustness of networks to such noisy labels in a "synthetic" experiment (much like Zhang et al. where labels are "synthetically perturbed") where we have access to the noisy labels. We show that in this scenario, LOO enables a faithful estimation of the true performance and analyzing the resulting expression for LOO might lead to insights regarding generalization for different types of noise settings. We will make sure to highlight this distinction, thank you for pointing this out.
> >
> >
> > *No test loss for transfer learning, why random initialization?*
> > In the context of deep networks, accuracy is the predominant metric and usually easier to interpret. For completeness we also added the test loss and LOO loss values. Due to space constraints we moved the table to the appendix, you can find it in Section C.4. We added the random initialization experiment to highlight that transfer learning does have a beneficial effect on the performance, implying that the feature map did adapt to the data to a strong degree. This makes it more surprising that the LOO estimates are still very faithful. We will update the text to point this out.

---

### Official Review · Reviewer_6oHc · 2021-11-02

**Correctness:** 3
**Technical Novelty And Significance:** 3
**Empirical Novelty And Significance:** 2
**Recommendation:** 6
**Confidence:** 4

**Main Review:**

Strength:

- Understanding generalization in deep learning remains one of the most important questions in deep learning theory. Studying the generalization ability of deep neural networks through LOO error is an interesting direction opened up by correspondence to kernel methods. As authors state, the study of LOO error in deep learning is quite underexplored and could be a promising direction.

- Demonstrated agreement is quite good and indicates LOO metrics can be useful in kernel regression.

- The paper is clearly written with good organization of presentation.

Weakness:

- There are many other generalization measures that could be compared to LOO error. Among all of them, why is LOO metrics preferred?This is not clearly answered since there’s no comparison. For example, pure validation split which is used in common practice (e.g. set aside 10% of the MNIST/CIFAR-10 training set). I believe all the phenomena observed by the authors could also be obtained using this metric without extra compute burden vs LOO (both still require O(n^3) computation for inference and require O(n^2) kernel element computations). In that case why is LOO a more interesting metric? (This can also extend to k-fold CV (in this case computation can be more expensive), Generalized cross validation, KARE (Jacot et al., NeurIPS 2020) …

- Another weakness is that the author is ignoring the computational cost of kernel computation. While FC based kernels used in the experiments often have negligible compute cost to build kernel K, for more realistic convolutional kernel with pooling would require O(100)- O(1000) GPU hours for CIFAR-10, and for the examples ResNet-152 on ImageNet would be naively 30k times more considering O(d^2 N^2) scaling which is still inefficient. Moreover, considering O(N^3) computation needed for computing A, the example case of estimating LOO error is challenging in the kernel case as well.

- One aspect I’m struggling to see is whether the observed agreement is surprising or more-or-less expected.  One source of uncertainty is that, as far as the reviewer can tell, test performance agreement comparison is made on the “kernel” side not necessarily on the finite width NN. All the good empirical agreement between A_{Test} and A_{LOO} just means that for doing ridge regression with neural kernels LOO estimates are good estimators for test performance. Predicting generalization performance of deep neural networks still hinges on test prediction with kernels matching finite networks. This works for some idealized cases but one needs to be subtle as demonstrated in [Lee et al., NeurIPS 2020].

- To me, first contribution regarding multi-class setting expression is a straightforward application of well known LOO prediction in kernel regression to the setting of multi-class classification as regression thus does not seems sufficiently significant.

- The paper fails to cite the main library used for empirical evaluations: (Neural Tangents library: [Novak et al., ICLR 2020]). Few other references should be added (suggested below in corrections)

Corrections:

- In related work, for the GP correspondence, [Lee et al., 2018] should be co-cited with [Matthews et al., ICLR 2018] and [Neal 1994] (for a single hidden layer).
- Also for extension to various architectures consider citing ([Garriga-Alonso et al., ICLR 2019], [Novak et al., ICLR 2019] for Conv and [Hron et al., ICML 2020] for self-attention and [Yang 2019, 2020] with Tensor programs)
-In section 3: treating classification as a regression task for common benchmarks for infinite width kernels started in [Lee et al., 2018] where they provide justification for doing it. See also [Hui & Belkin ICLR 2021] for using squared loss in practical DL workloads.
- Why binarize the MNIST dataset in section 4.3? (meant two digit (binary) classification? Inferring from 0.5 accuracy): note binarized MNIST often means that input pixels are binarized to 0/1.


Nit:
- Please, include references in the main submission. Instructions for ICLR 2022 submission were to include both references and the appendix in the main file.

-----------------------------------------------
Post rebuttal: Thank you for the response. I've read the clarification by the authors and interaction with other reviewers. With some of the improvements, I'm ok to support acceptance based on authors showing interesting generalization phenomena in DL using LOO error on kernels. I've raise my score.

**Summary Of The Paper:**

This paper investigates leave-one-out (LOO) error as a generalization measure for wide, deep neural networks using the correspondence to the NTK.  The authors show LOO error also shows behaviors in DNN such as random label fitting, double descent and transfer learning.

Specific contributions claimed by the authors are:
- Extend LOO error for multi-class setting and closed-form formula for LOO accuracy
- Demonstrate empirically LOO loss and accuracy capture generalization ability of deep learning in variety of settings
- Utility of LOO on practical networks predicting transfer performance
- Utilize the mathematical form of LOO loss to derive insights into double descent and the role of regularization


**Summary Of The Review:**

This paper studies generalization of deep learning with LOO error using neural kernels. This is a relatively novel tool that has not been thoroughly explored in deep learning theory and opens up a new window of studying generalization properties of wide neural networks. There are still some questions regarding the effectiveness or usefulness of the proposed metric detailed above.  With current limitations, the paper is borderline and leaning rejection. However if authors could clarify some of the doubts, happy to move up the score for acceptance.

---

> ### Author Response · Authors · 2021-11-19
> **Answer to Reviewer 2 (I)**
>
> Thank you for your very detailed feedback! We are glad to hear that you enjoyed reading our paper. We will address your comments and questions in the following.
>
> *More generalization measures, why LOO:*
> Thank you for pointing this out! We agree with you that we should clearly explain this. There are two advantages of the leave-one-out error over other generalization measures: 1) The leave-one-out error only relies on the training data. From a practical perspective, this means that one does not have to put some part of the data aside for validation but one can rather use all the data for training. 2) A more significant benefit is in terms of the theoretical analysis. Mathematically studying a hold-out estimate is extremely difficult since we need to reason about predictions the model makes on datapoints that it has not seen during training. In contrast, the leave-one-out error only involves quantities evaluated on the training data, for which we can do direct analytical computations. This is demonstrated through our double descent analysis, where the LOO error allows for a simple and rather intuitive proof for the spike occurring at the interpolation threshold. A similar analysis for a held-out test set seems intractable on the other hand (to the best of our knowledge). K-fold validation also only involves the training data, but, interestingly, it does not allow for a closed-from solution as LOO does. While repeating the training of a model K times might be feasible for small K, calculating LOO is certainly simpler and theoretically way more attractive due to the closed-form formula, as we argued above. Generalized cross validation is an interesting suggestion as it constitutes an approximation to the LOO. The approximation is mainly motivated through computational reasons and expected to lead to an inferior approximation for small sample sizes. Since the computational costs were not heavy in the scenarios we considered, we used LOO instead. Generalized cross validation might be a valid alternative however for very large sample sizes. We will update the text accordingly.
>
> *Computational costs of kernels:*
> We agree with this observation, neural kernels do scale poorly as soon as convolutional layers and pooling are involved. This is however a limitation of kernels themselves since basic inference (i.e. predicting on unseen samples) also incurs the same computation limitations (in fact, computing the validation loss is even worse since one needs to build the train-test kernel as well). We thus think that this is not necessarily an argument against the usage of the leave-one-out error itself but rather a limitation of the corresponding kernels. To better answer your comment, we also would like to emphasize that the aim of this work is not to advocate that leave-one-out error is a way to estimate the performance of state-of-the-art networks, or their corresponding kernels, on huge datasets. Instead, we demonstrate that the LOO serves as a tool that captures a variety of phenomena exhibited by deep models, which are sometimes not fully understood, even for fully-connected networks (whose kernel is efficiently calculatable). We do however also demonstrate the utility of LOO for modern networks by studying a transfer learning setting, where we can avoid large computational costs.
> To address your comment, we will explicitly mention the cost (along with appropriate references) and clarify the utility of the LOO as an analysis tool. Further comments are appreciated.
>
> *Infinite vs finite width networks:*
> This is correct, we estimate the generalization performance of the underlying infinitely-wide network. We did not perform experiments for finite width networks since there is a large body of theoretical work, proving how very wide networks become essentially indistinguishable from their kernel counterpart [Arora et al., 2019, Lee et al., 2019]. As you point out however, there is a gap in performance to networks out of the kernel regime and indeed our analysis cannot explain their behaviour. We apologize if our writing suggested otherwise. We would however like to remark that understanding the generalization of such wide networks is still far from complete and LOO could serve as a great tool to advance this understanding.
>
> *New LOO formulas are straightforward:*
> Yes, we agree with you that the extension of the regularized leave-one-out error to the multi-class setting and accuracy is relatively straightforward. On the other hand, this extension introduces a more natural measure for tasks encountered in deep learning and is hence important for our work. Moreover, as Reviewer 1 points out, the case of zero regularization is rather uncommon in the statistics literature and we also present extensions to this case as well.

---

> > ### Author Response · Authors · 2021-11-19
> > **Answer to Reviewer 2 (II)**
> >
> > *Missing citation for neural-tangents:*
> > We indeed forgot to add this citation. We heavily rely on both Jax and the neural tangents library for our experiments, and should indeed give them the credits they deserve. We added both citations to the main text.
> >
> > *Additional citations:*
> > Thank you for pointing out those additional works. We have added their citations to the related works.

---

### Official Review · Reviewer_k266 · 2021-11-02

**Correctness:** 4
**Technical Novelty And Significance:** 3
**Empirical Novelty And Significance:** 2
**Recommendation:** 6
**Confidence:** 3

**Details Of Ethics Concerns:**

I did not spot any signs of ethical issues.

**Main Review:**

The main novelty of the manuscript is the characterization of the double descent behavior of ridge (or ridgeless) regression based multi-class classification in terms of its closed form leave-one-out formulations and spectral decomposition of the kernel matrix. This provides a simple and intuitive insight to this phenomenon, and demonstrates how the leave-one-out error spikes at interpolation point with quite mild assumptions.

The various forms of leave-one-out computations for ridge regression are quite well known and therefore do not provide that much novelty, see
http://128.30.100.62:8080/media/fb/ps/MIT-CSAIL-TR-2007-025.pdf
for a summary and especially the multi-class classification has been implemented in several software libraries such as
https://citeseerx.ist.psu.edu/viewdoc/download?doi=10.1.1.676.9100&rep=rep1&type=pdf
and
http://jmlr.org/papers/v17/16-470.html
The details concerning the "ridgeless" case may be less well known.

While the authors intent is to connect the analysis to the recent deep neural networks research, it makes the overall story somewhat confusing, and focusing to the leave-one-out computations of ridge regression only, without NTK etc., would have made considerably sharper message. In addition, the results should be put into context of the other results concerning linear regression presented by Bartlett et al. (2020), Mei and Montanari (2021), etc. that the authors already cite.


**Summary Of The Paper:**

The manuscript considers the use leave-one-out error to analyse the generalization ability of one-versus-all multi-class classification models that can be expressed as multiple-output ridge regression models. As examples of such models, the authors review certain variations of neural networks. Specific attention is given on the double descent phenomenon showing that the traditionally considered U-shaped curve demonstrating the trade-off between underfitting and overfitting areas can be extended over the interpolation point after which the generalization ability again tends to improve. The spiking effect of the generalization error is derived from the closed form solutions of the leave-one-out computations for ridge regression based methods.


**Summary Of The Review:**

The mathematics is clear and convenient to read but the overall story is somewhat confusing due to the connections to neural networks and some other branches that do not contribute too much to the main message. The closed form solutions of the leave-one-out computations are not very novel and the most can be straightforwardly derived from the existing literature. I find the idea of considering the double descent behavior emerges from the closed form leave-one-out formulations to be interesting and it may deserve more attention.

---

> ### Author Response · Authors · 2021-11-19
> **Answer to Reviewer 1**
>
> Thank you for the insightful feedback! We are happy to hear that you liked our analysis of double descent. We address your comments and questions below.
>
> *Novelty of LOO extensions:*
> We agree that the extension of the regularized leave-one-out error to the multi-class setting is a simple extension of the standard leave-one-out error. We were not aware that other software packages have implemented this, and will gladly point this out. While the extension to accuracy is not complicated, it was never presented in the literature and is a more natural measure for the scenarios usually considered in deep learning. Moreover, as you pointed out, the case of zero regularization is less common and essential to the scenario we study. We have reformulated our claims accordingly and cited the mentioned work.
>
> *Kernels vs Neural networks:*
> We apologize for the confusion regarding the connection of the leave-one-out error and neural networks. The aim of our work is to leverage the fact that infinitely-wide neural networks correspond to kernels, thus making the leave-one-out error tractable in the context of very wide networks. This is the reason why we investigate the LOO error in typical deep learning scenarios. We will update the text to clearly distinguish kernels and neural networks.
>
> *Discussion of Bartlett et al. and Montanari et al.*
> Thank you for pointing this out. Bartlett et al. study double descent in the context of linear regression. They assume a linear teacher model for the labels as well as that the inputs can be disentangled into independent sub-gaussian factors. Similarly, Montanari et al. also consider labels generated by a teacher and inputs are restricted to come from the uniform distribution on the sphere. The considered model is specialized to a random feature model. Our work on the other hand does not impose any assumption on the labels and only a weak assumption on the inputs indirectly by requiring that the eigenvectors of the Gram-matrix are uniformly distributed on the sphere. Moreover, we analyze the general kernel setting and not only linear regression or random feature regression. We want to highlight however that the main advantage of our analysis of the spike for double descent is its simplicity. The work of Montanari et al. for instance needs very involved tools from random matrix theory. The leave-one-out error on the other hand allows for a simple and intuitive argument. We will update the text accordingly.

---

> > ### Comment · Reviewer_k266 · 2021-11-29
> > **Acknowledgement of the answer**
> >
> > Dear authors, thank you for the detailed answers for my comments. The suggested fine tuning of the manuscript sounds reasonable.

---

### Decision · Program_Chairs · 2022-01-20

**Decision:**

Accept (Poster)

**Comment:**

This paper proposes to use LOO to characterize the generalization error of neural networks via the connection between NN and kernel learning. The reviewers find the new results interesting. The meta reviewer agrees and thus recommend acceptance.